# A mouse model of cardiac immunoglobulin light chain amyloidosis reveals insights into tissue accumulation and toxicity of amyloid fibrils

Gemma Martinez-Rivas [1,2], Maria Victoria Ayala[1,2], Sebastien Bender[1,2], Gilles Roussine Codo[1,2], Weronika Karolina Swiderska[1,2], Alessio Lampis [1,2], Laura Pedroza[3], Melisa Merdanovic[3], Pierre Sicard[4], Emilie Pinault [5], Laurence Richard[6], Francesca Lavatelli [7,8], Sofia Giorgetti[7,8], Diana Canetti[9], Alexa Rinsant[10], Sihem Kaaki[10], Cécile Ory[10], Christelle Oblet [1], Justine Pollet [1], Eyad Naser[11], Alexander Carpinteiro[11], Muriel Roussel[1,2], Vincent Javaugue [1,2,12], Arnaud Jaccard [1,2], Amélie Bonaud[1,2], Laurent Delpy [1,2], Michael Ehrmann [3], Frank Bridoux[1,2,12] & Christophe Sirac [1,2] ✉

Immunoglobulin light chain (LC) amyloidosis (AL) is one of the most common types of systemic amyloidosis but there is no reliable in vivo model for better understanding this disease. Here, we develop a transgenic mouse model producing a human AL LC. We show that the soluble full length LC is not toxic but a single injection of pre-formed amyloid fibrils or an unstable fragment of the LC leads to systemic amyloid deposits associated with early cardiac dysfunction. AL fibrils in mice are highly similar to that of human, arguing for a conserved mechanism of amyloid fibrils formation. Overall, this transgenic mice closely reproduces human cardiac AL amyloidosis and shows that a partial degradation of the LC is likely to initiate the formation of amyloid fibrils in vivo, which in turn leads to cardiac dysfunction. This is a valuable model for research on AL amyloidosis and preclinical evaluation of new therapies.

Systemic light chain (AL) amyloidosis is a rare but severe acquired protein misfolding disease characterized by deposition of amyloid fibrils composed of a monoclonal immunoglobulin light chain (LC) produced in excess by a B or plasma cell clone[1]. The LCs involved in amyloidosis have a propensity to aggregate into the characteristic β-sheet structure of amyloid fibrils, and accumulate in the extracellular compartments of tissues, leading to organ dysfunction. Renal and cardiac manifestations are the most frequent, the latter being

[1]CNRS UMR7276/INSERM U1262, University of Limoges, CRIBL lab, team 3 BioPIC, Limoges, France. [2]French National Reference Centre for AL Amyloidosis and Other Monoclonal IG Deposition Diseases, University Hospital, Limoges, France. [3]University Duisburg-Essen, Centre for Medical Biotechnology, Essen, Germany. [4]PhyMedExp, IPAM/Biocampus (IBiSa/France Life Imaging), UMR INSERM 1046-CNRS 9214, universityof Montpellier, Montpellier, France. [5]BISCEm (Biologie Intégrative Santé Chimie Environnement) Platform, US 42 INSERM/UAR 2015 CNRS, University of Limoges, Limoges, France. [6]Department of Pathology, University Hospital, Limoges, France. [7]Department of Molecular Medicine, Institute of Biochemistry, University of Pavia, Pavia, Italy. [8]Research Area, Fondazione IRCCS Policlinico San Matteo, Pavia, Italy. [9]Centre for Amyloidosis, Division of Medicine, University College London, London, UK. [10]Department of Pathology, University Hospital, Poitiers, France. [11]Department of Hematology and Stem Cell Transplantation, West German Cancer Center, University Hospital Essen, Essen, Germany. [12]Department of Nephrology, University Hospital, Poitiers, France. ✉e-mail: christophe.sirac@unilim.fr

associated with poor outcomes[2]. The molecular mechanisms that lead to the aggregation of LCs have been extensively studied, primarily in vitro due to the lack of other models reproducing the early stages of the disease. The variable domain (VL) of LCs has long been suspected to be the causative part of AL amyloid fibril formation since only a few VL germline genes account for most of the cases[3,4] and destabilizing mutations acquired during affinity maturation in the V domain have been shown critical for amyloidogenicity in vitro[5,6]. Consequently, our knowledge on LC aggregation process has been mostly obtained from isolated amyloidogenic VL since full-length LCs seem to be resistant to aggregation under physiological conditions[6–8]. Accordingly, cryo-electron microscopy (Cryo-EM) structures of ex vivo AL amyloid fibrils confirmed that the cross β-structured interactions within the core of the fibrils are primarily established by the VL domains, occasionally augmented by a few amino acids from the N-terminal part of the CL[9–11]. The remaining portion of the LC, comprising most of the constant domain (CL), seems to be disorganized outside the fibrillar structure and partially or totally cleaved[12]. This prompts the question of whether proteolysis in the CL is required to initiate amyloidosis formation or just a subsequent degradation of non-fibrillar parts of the LCs. The multiple cleavage pattern in sites not accessible in the native dimers suggests a fragmentation subsequent to aggregation[12]. However, the high amyloidogenicity of some fragmented species, as opposed to the stability of full-length LCs, suggests that proteolysis of the LCs could also be required to initiate amyloidosis formation[8].

In addition to the mechanical stresses caused by the accumulation of amyloid fibrils in tissues, the soluble form of LCs may also contribute to cardiac toxicity. Patients responding to treatments that aim at reducing circulating LCs show a significant decrease in NT-proBNP concentrations, correlated with improved cardiac function, in spite of the absence of a significant decrease in the amyloid burden[13]. Studies conducted in vitro using cardiomyocyte and cardiac fibroblast cultures, as well as in *C. elegans* and Zebrafish models, support this theory[14–18]. Exposure of these models to soluble amyloidogenic LCs leads to cellular stress through internalization of the LC increased production of reactive oxygen species (ROS) and the activation of a non-canonical MAPK pathway. As a result, lysosomal dysfunction, autophagy impairment, and mitochondrial damage were observed, followed by cell death. Although these studies provide valuable insight into LC toxicity on cardiac cells, they neither reproduce the cellular complexity, the microenvironment, and the shear stress observed in human tissue, nor the deposition and accumulation of amyloid material composed of the LCs. Translating these findings to human physiology remains challenging.

Several approaches to creating rodent models of AL amyloidosis have been attempted over the years[19]: Firstly, by the massive injection of Bence-Jones proteins, purified fibrils from patients or the so-called "amyloidoma" composed of a crude grinded human tissue containing amyloid material[20,21]. Although localized amyloid material can be observed in these models, their application is limited to therapeutic studies since they poorly reproduce the human pathophysiology and organ involvement. Lately, transgenic approaches to produce amyloidogenic LCs endogenously in rodents with mostly disappointing results[22,23]. One of them succeeded at reproducing amyloid deposition, even though deposits were localized in the stomach of aged mice, suggesting a destabilization of LCs in the local acidic environment which promoted aggregation rather than a physiological amyloid formation in classically involved organs[24]. This apparent resistance to amyloidosis in mice was also observed for other types of systemic and localized amyloidosis and was attributed to better proteostasis and a rapid turnover of proteins[25–27]. One of the main limits of these models is the levels of circulating free LCs (fLC) which are much lower than those observed in patients, likely failing to reach the threshold needed to initiate fibril formation[28]. To overcome this limit, we have developed a unique transgenic approach

that allows to achieve high levels of circulating pathological LCs into the mice[29]. This strategy has been successful for modeling non-amyloid monoclonal LC-related deposition diseases affecting the kidneys, including light chain deposition disease and light chain renal Fanconi syndrome[30,31].

In the present study, we applied this transgenic approach to create a mouse model producing an AL amyloidosis LC to overcome the lack of in vivo models for studying this devastating disease and provide a reliable tool for preclinical investigations of new therapies or diagnostics. Mice present with a high level of free amyloidogenic LCs but only exceptionally develop spontaneous systemic AL amyloidosis. However, upon induction not only with preformed fibrils but also with soluble amyloidogenic VL, they develop robust amyloid deposits in the heart, spleen, vessels, and to a lesser extent in the kidney and other organs. This study characterizes this new AL mouse model and offers new insights into the pathophysiological mechanisms of AL amyloidosis in vivo.

## Results

### High levels of circulating amyloidogenic lambda-free light chains in mice are not enough per se to induce AL amyloidosis or symptomatic toxicity

To generate the mouse model of AL amyloidosis, we used the cDNA coding a monoclonal λ-LC derived from a patient with renal and cardiac AL amyloidosis (λS-PT) (Fig. S1A). λ-fLC level at diagnosis was 128 mg/L with a dfLC at 110 mg/L. The monoclonal LC (λS-LC) sequence is derived from an *IGLV6-57* variable germline gene, the most frequent in AL amyloidosis[32], rearranged to an *IGLJ3* junction segment and an *IGLC3* constant domain. Mutation analysis revealed 14 mutations compared to the VJ germline sequences (Fig. S1B) with three of them resulting in an aggregation-prone region exposed to the solvent (Fig. S1C).

High production of the pathogenic human free λS-LC in the mouse was achieved using our original transgenic strategy, associating targeted insertion at the *Ig kappa* locus of the human *λS-LC* gene and backcrossing with the DH-LMP2A strain (thereafter called DH)[33] that allows a normal B cell development with an increased plasma cell number in the absence of Ig heavy chains (Fig. 1A)[34]. DH mice produce ~100 μg/mL of polyclonal mouse free κ-LCs[31] and were used as controls throughout the study except when otherwise stated. As expected, the resulting double homozygous λS-DH mice presented with a high proportion of spleen PCs (8.9% ± 3.9, $n = 3$) mostly producing the human transgenic λS-LC (76.3% ± 2.7), mimicking features of monoclonal gammopathies (Fig. 1B).

FLC serum level in λS-DH double homozygous mice reached 496 ± 41 μg/ml (mean ± SEM, $n = 20$) in 2–6 month-old mice and remained stable in older mice (499 ± 46 μg/ml, 6–18 month-old, $n = 16$), which is 4 fold more than in the patient λS-PT at diagnosis (Fig. 1C). Serum fLC level in the λS mice (without DH backcrossing) was only 37.2 μg/ml ±5.6 (mean ± SEM, $n = 4$), likely due to the association of the human LC to mouse HC, forming full Igs[30]. Western blot using a mouse anti-human IGLV6 antibody confirmed the presence of the human λS-LC in the sera of λS-DH mice (Fig. S1D).

Human LCs were readily detectable by immunofluorescence in the spleen, especially in extrafollicular plasma cells and in proximal tubular cells of the kidney, corresponding to the physiological tubular reabsorption of the LC (Fig. S1E). However, we did not observe any amyloid deposits in 3–12-month-old mice ($n = 21$). λS-DH mice had a normal lifespan compared to control DH mice with no signs of morbidity. Serum creatinine and albuminuria levels at 8 months old were 4.40 μM/L ± 1.51 (mean ± SEM, $n = 5$), and 43.93 μg/ml ± 9.54 ($n = 6$), respectively, which is comparable to previously published data in DH and WT mice[31]. Altogether, these results show that the transgenic amyloidogenic LC does not seem to display apparent direct toxicity or have a global negative effect on the survival of mice.

## Variable domain of the λS-LC is amyloidogenic in vitro

In order to understand why λS-DH mice do not develop AL amyloidosis, we sought to characterize the amyloidogenicity of the λS-LC and its VL domain in vitro. We produced both recombinant forms, namely the Full-Length LC (rλS-LC) and its variable domain alone (rλS-VL). Corresponding amino acid sequences and alignment are depicted in Fig. S2A. As expected, rλS-LC mainly constituted dimers that were stabilized by the disulfide bond between constant regions and rλS-VL remained under its monomeric form (Fig. S2B, C). The melting temperature ($T_m$) of the rλS-VL protein was 38.7 °C (Fig. 2A), which is similar to Wil, another amyloidogenic *IGLV6-57*−derived VL widely used for in vitro aggregation studies[35]. Conversely, the rλS-LC protein had two $T_m$ at 48.5 and 59.4 °C (Fig. 2A), likely corresponding to the minor monomeric form and the dimeric form respectively. Accordingly, rλS-VL started fibrillogenesis in vitro at 2 days under aggregating conditions as observed with the increase of ThT fluorescence, while the rλS-LC remained at baseline throughout the experiment (214 h) (Fig. 2B). Seeding with sonicated preformed fibrils (seeds) highly increased the kinetic of fibrillogenesis of rλS-VL, reaching the plateau phase after a few hours, but not rλS-LC, which remained at baseline (Fig. 2B). Electron microscopy confirmed the presence of fibrils formed by rλS-VL but not the full-length rλS-LC (Fig. 2C).

Altogether, these results confirm that the full-length λS-LC seems to be resistant to fibril formation in vitro, which corroborates the in vivo findings in the λS-DH mice. However, the λS variable domain is able to form fibrils in vitro.

## Seeding with recombinant λS-VL fibrils induces fast AL amyloid deposition in vivo

Seeding with ex vivo material (Amyloid Enhancing Factor, or purified fibrils from a human or mouse) was necessary to induce or accelerate amyloidosis formation in other animal models of AA and ATTR[36,37]. Unfortunately, we did not have access to tissue from the patient (λS-PT) but we hypothesized that the rλS-VL fibrils made in vitro could serve as seeds for the elongation of amyloid fibrils with the endogenous λS-LC in our transgenic mice (Fig. 3A). Accordingly, intravenous injection of in vitro rλS-VL sonicated fibrils resulted in the formation of amyloid deposits in organs, starting as soon as 48 h after the injection. Congo Red (CR) birefringence and fluorescence showed that cardiac and surrounding vascular tissues were principally affected together with the spleen (Fig. 3B). In contrast to the patient λS-PT, λS-DH mice rarely developed glomerular kidney deposits, only visible in mice with high deposits in heart and spleen (Fig. 3B). Cardiac deposits were found mainly in the myocardial compartment of the ventricular and atrial walls, along the muscle fibers, resulting in diffuse thin filaments outlining the cells and in cardiac blood vessels (Fig. 3C). Not all seeded mice developed amyloidosis but the penetrance increase with time, ranging from 28% one week after injection to 100% after 6 months (Fig. 3D). A scoring system was established according to the CR staining in the heart (Fig. S3A). Using this score, we showed that there is a high heterogeneity of amyloid burden in the positive mice whatever the time after induction is (mean scores -1.5 with range 1−3), except for long-term inductions which presented with a higher average score (>6 months; mean score 2.67, range = 1−3, n = 6) (Fig. 3E). We did

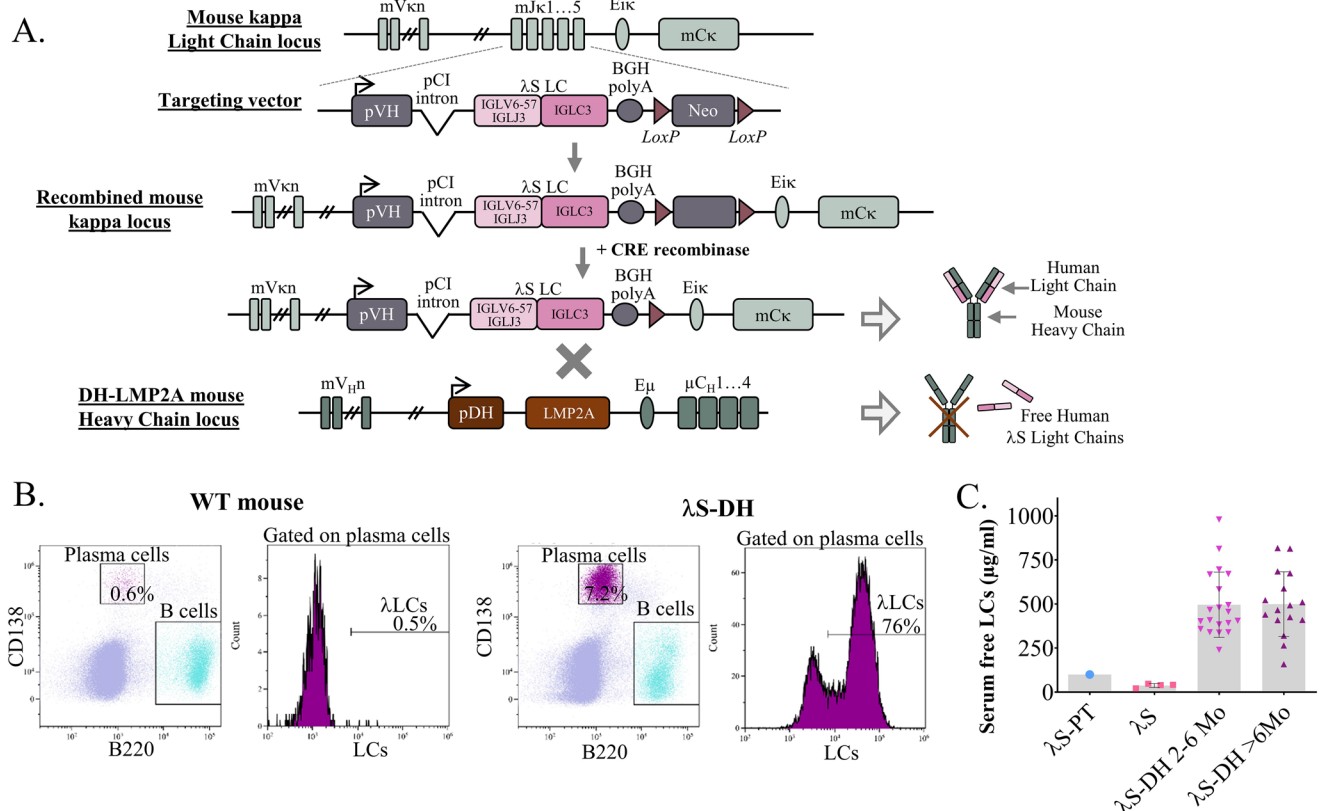

**Fig. 1 | Generation of the transgenic mouse model λS-DH. A** Transgenic strategy used to achieve a high production of human free light chains in the mice consists in the insertion of the λS-LC from the λS-PT in the mouse κLC locus. These mice were backcrossed with the DH-LMP2A mice to avoid the association between human LCs and mouse-heavy chains. **B** Flow cytometry on spleen cells of WT and λS-DH mice. CD138 and B220 staining allows to identify plasma cells and B cells respectively. Intracellular staining of with anti hλ-LC allows to show the plasma cells producing the transgenic λS-LC. Graphs are representative examples from the analyzed mice (n = 3 WT and n = 3 λS-DH). **C** Serum dosage of the circulating free light chains in the patient (λS-PT) at diagnosis and the transgenic mice without (λS, n = 4) and with (λS-DH) the DH-LMP2A allele of 2–6 months old (n = 20) and >6 months old (n = 16) mice. Representation of single values with mean (gray bar) ± SD.

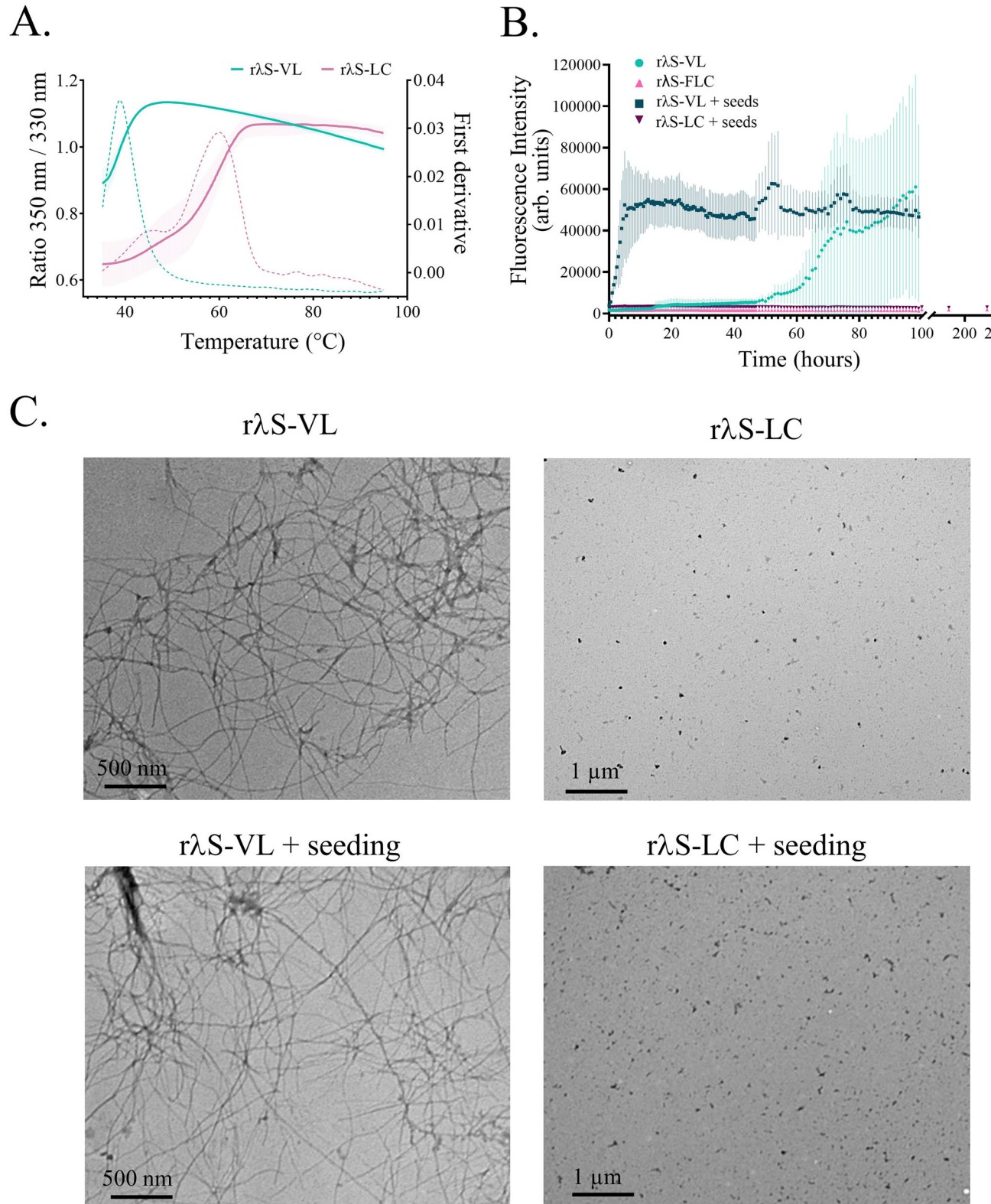

**Fig. 2 | In vitro characteristics of λS-LC species. A** Stability of the recombinant λS proteins, rλS-LC (pink) and rλS-VL (green) was measured by the ratio of the fluorescence at 350 and 330 nm (solid line) upon thermal denaturation. The melting temperature (Tm) for each sample was given by the maximum peaks of the 350/330 nm ratio first derivative (dotted line). The curves correspond to the mean of 3 different measurements for the rλS-LC and 4 for the rλS-VL, and the error bars correspond to SD. **B** Kinetics of fibril formation in vitro with the different λS protein species with (light green and pink) or without seeding (dark green and pink). The aggregation kinetics was followed by the measurement of the fluorescence of Thioflavin T. Represented in mean ± SD of 3 different experiments in duplicate. **C** Representative TEM analysis of the samples from (**B**) without (upper panel) or with (lower panel) seeding.

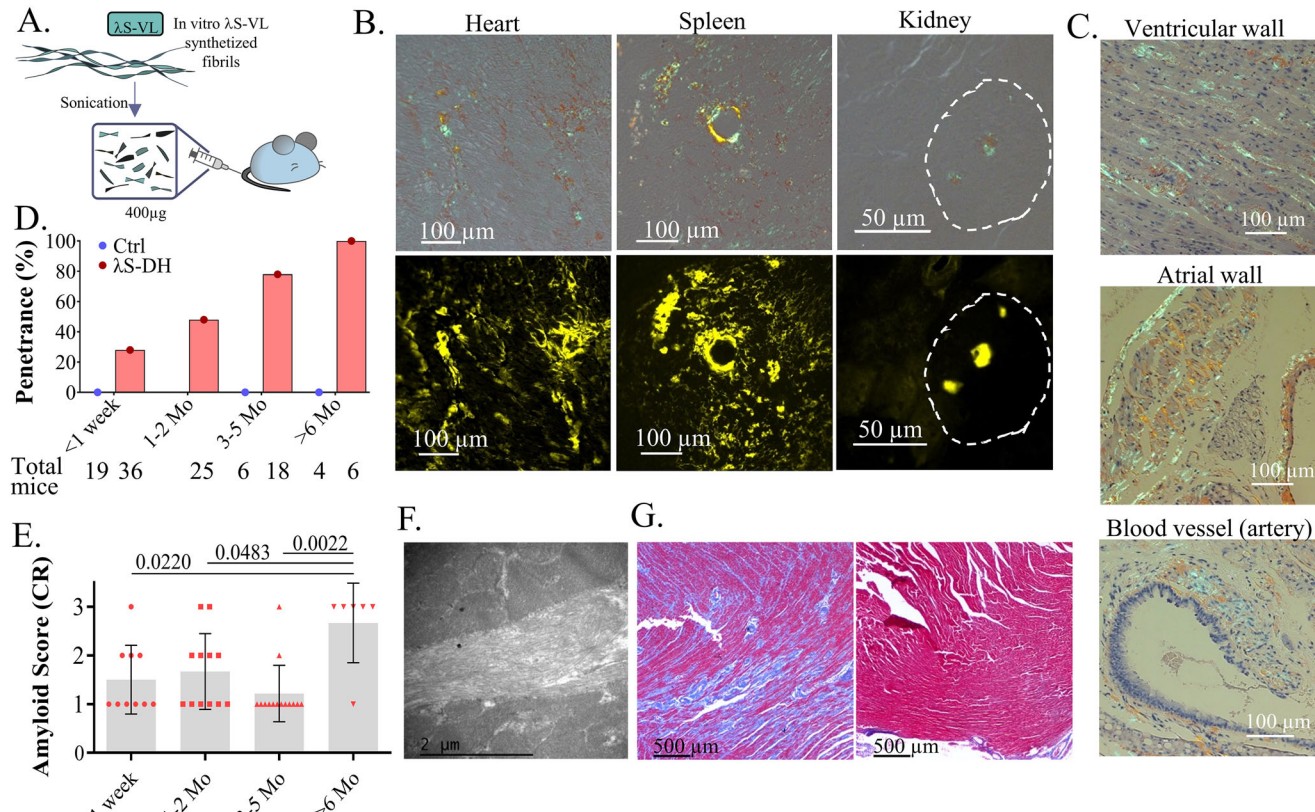

**Fig. 3 | Amyloidosis induction in λS-DH mice with VL fibrils. A** Protocol of amyloidosis induction with rλS-VL seeds. **B** Histological characterization of CR birefringence under polarized light (top) and fluorescence (bottom) on frozen tissues. A glomerulus is circled with dotted line. Representative example of a mouse analyzed at 6 months after induction. **C** Hematoxylin-Eosin and CR staining in a paraffin-embedded heart. Birefringence was visible in the myocardium of the ventricular wall, atrial wall, and around blood vessels throughout the tunica media and adventitia. Representative example of a mouse at 9 months after induction. **D** Organs from induced mice (λS-DH, red, and controls, blue) were analyzed histologically with CR staining at different time points after the injection. AL-positive mice were determined by the presence of CR fluorescence in the heart, and the penetrance was calculated as the ratio of positive mice to the total number of mice analyzed at each timepoint (indicated below the graph) (**E**) The amyloidosis score was calculated for the positive mice from (**D**) ($n = 10$ at <1 week, $n = 12$ at 1–2 months (Mo), $n = 14$ at 3–5 months and $n = 6$ at >6 months) by the CR fluorescence in the cardiac tissue, corresponding to: score 1 (low), score 2 (mid) and score 3 (high). Score assessment is described in the methods and Fig. S3A. Representation of the single values with mean (gray bar) ± SD. Comparisons were performed by Mann–Whitney two-sided test. Exact *P* values are indicated. **F** Electron microscopy of the cardiac tissue showing the amyloid fibrils in the extracellular compartment of a score 3 mouse. **G** Representative image of fibrotic tissue revealed with a Masson's Trichrome staining in the hearts of a score 3 mouse (left) and a control (right).

not detect amyloid deposits in control mice injected with the same amount of in vitro fibrils, and either producing polyclonal murine LCs (DH, $n = 7$), a human monoclonal non-amyloidogenic κLC (κR-DH, $n = 13$) or full polyclonal Ig (WT, $n = 9$) (Figs. 3D and S3B). This result ruled out the possibility that amyloid deposits detected in organs were due to the injected fibrils themselves. Ultrastructural studies of the cardiac deposits showed the typical conformation of amyloid fibrils, as unbranched, long filaments intertwined in the extracellular compartment of the tissues (Fig. 3F). In high-score mice, we observed myocardial interstitial fibrosis, a typical observation in patients (Fig. 3G), but no obvious immune cell infiltration (Fig. S3C)[38].

Immunofluorescence typing with a monoclonal anti-human λ-LC antibody recognizing specifically the constant region of the human λ-LCs (Fig. S3D) confirmed that amyloid deposits are made up of the transgenic λS-LC and not only the injected VL fibrils and mostly colocalized with CR positive areas in heart (Fig. 4A), spleen and kidney (Fig. 4B). Interestingly, in hearts with low amyloid burden, we readily detected human λ-LCs staining that not only colocalized with CR clusters but also with CR negative intercellular spaces (Fig. S3E). Such staining was never detectable in non-induced mice or control-induced mice (Fig. S3F). This suggests that prefibrillar aggregates or non-mature amyloid fibrils, not yet detectable by CR staining, are likely present in the tissue. Systemic AL deposits were also found in different amounts in the tongue, liver, lung, and visceral fat (Fig. S3G). Staining for APOA-II or SAA, two types of amyloidosis that can develop spontaneously in mice, was negative (Fig. S3H). Immuno-electron microscopy confirmed the AL typing, showing a strong binding of the gold-labeled anti-human λ-LC to amyloid fibrils but not with anti-human κ-LC (Fig. 4C). Altogether, these results show that the transgenic full-length λS-LC is an abundant constituent of amyloid fibrils in the λS-DH mice and can therefore elongate amyloid seeds composed of the VL alone to form mature amyloid deposits.

## Soluble amyloidogenic fragment of the λS-LC is able to induce amyloid deposition in λS-DH mice

We next sought to determine if the native soluble VL domain of the λS-LC, shown to be amyloidogenic in vitro, would also be able to initiate new fibrils in vivo. We injected the rλS-VL fragment previously used for in vitro fibril formation. To rule out the possibility that rλS-VL samples already contained aggregates prior to injection, we conducted several experiments including ThT assays, gel electrophoresis, and HPLC that all argued for the absence of detectable aggregates (Fig. S2B). A single intravenous injection of 800 μg of soluble rλS-VL to the mice (Fig. 5A) led to amyloid deposits starting at 2 months after the injection in few mice with increased penetrance at later time points (Fig. 5B). None of the control mice developed amyloidosis at 4 ($n = 4$) or 6 ($n = 4$) months

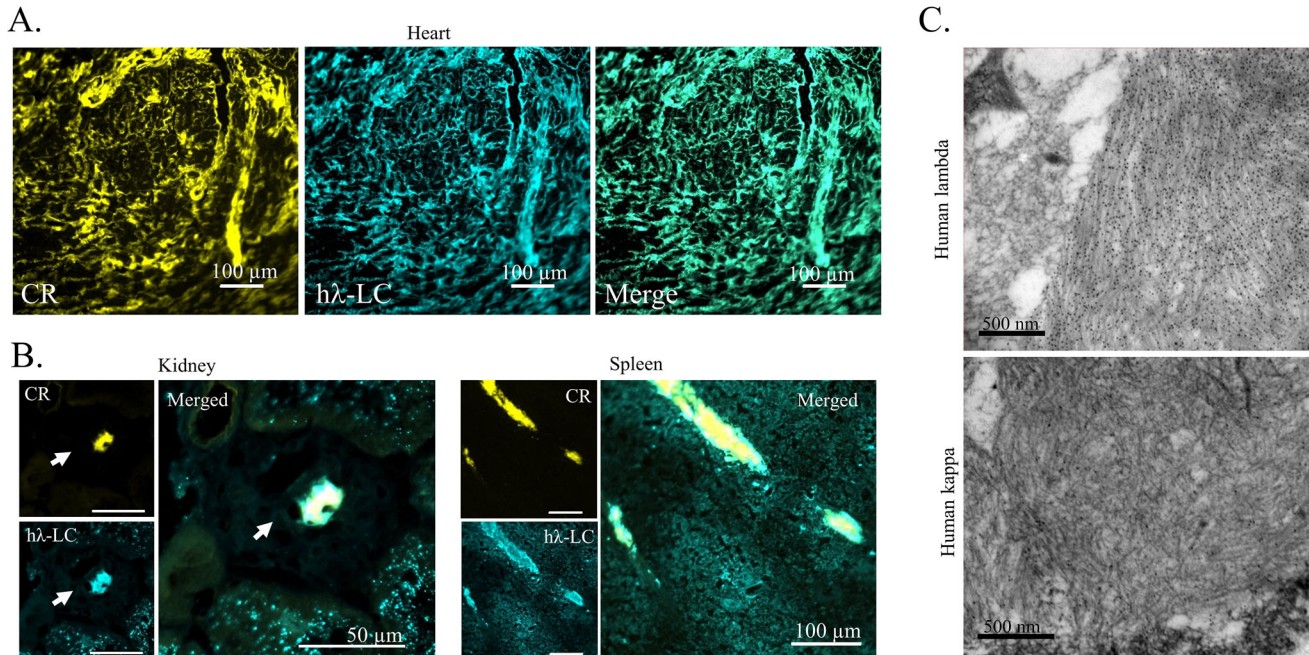

**Fig. 4 | Typing of the amyloid deposits in *λ*S-DH mice. A** Cardiac deposits were revealed by CR fluorescence and its colocalization with an anti-human *λ*LCs antibody coupled to FITC. This monoclonal antibody recognizes specifically an epitope of the constant domain of *λ*-LCs. Representative image of a score 3 mouse. All CR-analyzed mice were co-stained with this antibody. **B** Colocalization between CR staining and human *λ*-LCs was also confirmed in the kidney and the spleen of mice. Same mouse as in (**A**). **C** The typing of the amyloidosis was confirmed by immunoelectron microscopy (*n* = 2) with anti-human *λ*-LCs gold-labeled antibody (top) in the cardiac tissue of a score 3 mouse. Anti-human κ-LCs gold-labeled antibody was used as a control (bottom).

post-injection (Fig. 5B). Similarly to fibrils injections, the penetrance did not reach 100% and there was a heterogeneity in amyloid scores even 6 months after the initial injection of VLs (Fig. 5C). However, the general pattern of deposits and the organ tropism was the same as with fibrils injection (Fig. 5D). Although these results indicate that the *r*λS-VL fragments are likely initiating the first fibrillar aggregates, typing with the anti-*λ* constant region antibody confirmed that the soluble full-length transgenic LCs participated in their elongation and maturation (Fig. 5D). Altogether, these data demonstrate that few unstable fragments of the LCs can rapidly form stable aggregates in vivo that remain in the tissues and are sufficient to initiate elongation of AL amyloid fibrils.

**Long-term follow-up revealed exceptional spontaneous AL amyloidosis development with age**
In humans, AL amyloidosis can take several years to develop[39]. We also showed that when seeded with amyloidogenic VLs, detectable amyloid deposits often occur after several months in the mice, corroborating findings in other mouse models of systemic amyloidosis[25,37]. Consequently, we conducted a long-term follow-up on 47 *λ*S-DH mice. Since we never detected any sign of morbidity and we did not find at this time any biomarker that could help to detect the onset of the disease, we randomly sacrificed mice at 12–16 months (*n* = 16), 18–24 months (*n* = 21) and >24 months (*n* = 10). Two out of 47 mice developed AL amyloidosis at the age of 14 and 18 months (Fig. S4). This indicates that spontaneous amyloidosis may rarely happen in these mice but is not related to age.

**Deep characterization of amyloid deposits shows striking similarities with those from AL patients**
Purified fibrils from cardiac tissue of positive mice (Fig. S5A) contained mainly a ~11 kDa fragment recognized by the anti-IGLV6 antibody (Fig. 6A), slightly shorter than the full *r*λS-VL. Mass spectrometry analysis of purified fibrils readily detected the *λ*S-LC,

with peptides both from the VL and from the CL parts of the protein, together with common amyloid-associated proteins, such as ApoA4, ApoE and vitronectin[40,41] but Serum Amyloid P component (SAP) was barely detected in only 1/3 extracts (Fig. 6B). Accordingly, we were not able to detect SAP in cardiac tissues of amyloid positive mice by immunofluorescence (Fig. S5B) which could be due to the low level of circulating mouse SAP in our mice (Fig. S5C, 1.92 µg/ml ± 0.65, mean ± SEM, *n* = 12) compared to human (~25 µg/mL)[42]. We also detected Collagen VI subunits that have been recently described to be structurally associated with amyloid fibrils in a patient[11]. Then, we used a procedure for gentle enrichment of insoluble content in the presence of a protease inhibitor cocktail to avoid tryptic activity of collagenase that can degrade parts of the LCs not involved in the core of the fibrils[43]. Three AL-positive mice and two amyloid-negative samples (one non-induced *λ*S-DH mouse and one DH) were analyzed. Immunoblotting showed no immunoreactive band in the extracts from amyloid-negative mice, while multiple ones were visible in the amyloid-positive animals (~from 25 to 12 kDa) and consistent with the presence of the full-length LC and fragments (Fig. S5D). 2D-PAGE and 2D western blotting on amyloid-positive samples showed a strikingly similar fragmentation pattern of the *λ*S-LC in all the mice tested, but also to the one previously observed in the heart of a patient with another IGLV6-57-derived AL amyloidosis (AL55) (Fig. 6C and S5E, F)[12]. LC-MS/MS analysis of the main blotted spots excised from the Commassie-stained gel showed that the 25 kDa spots contained tryptic peptides from the entire LC sequence, except the last eight C-terminal amino acids which were not detectable (Fig. 6D). The lower spots, instead, contained the entire VL and progressively shorter portions of the constant domain. Some spots with MW < 12 kDa, not detectable by WB, were shown to contain exclusively the VL or portions of it. Taken together, our results confirm that the VL is the major common component of the AL amyloid deposits but that endogenous full-length *λ*S-LC participates in the formation of the insoluble material extracted from the tissue. The similarities in the

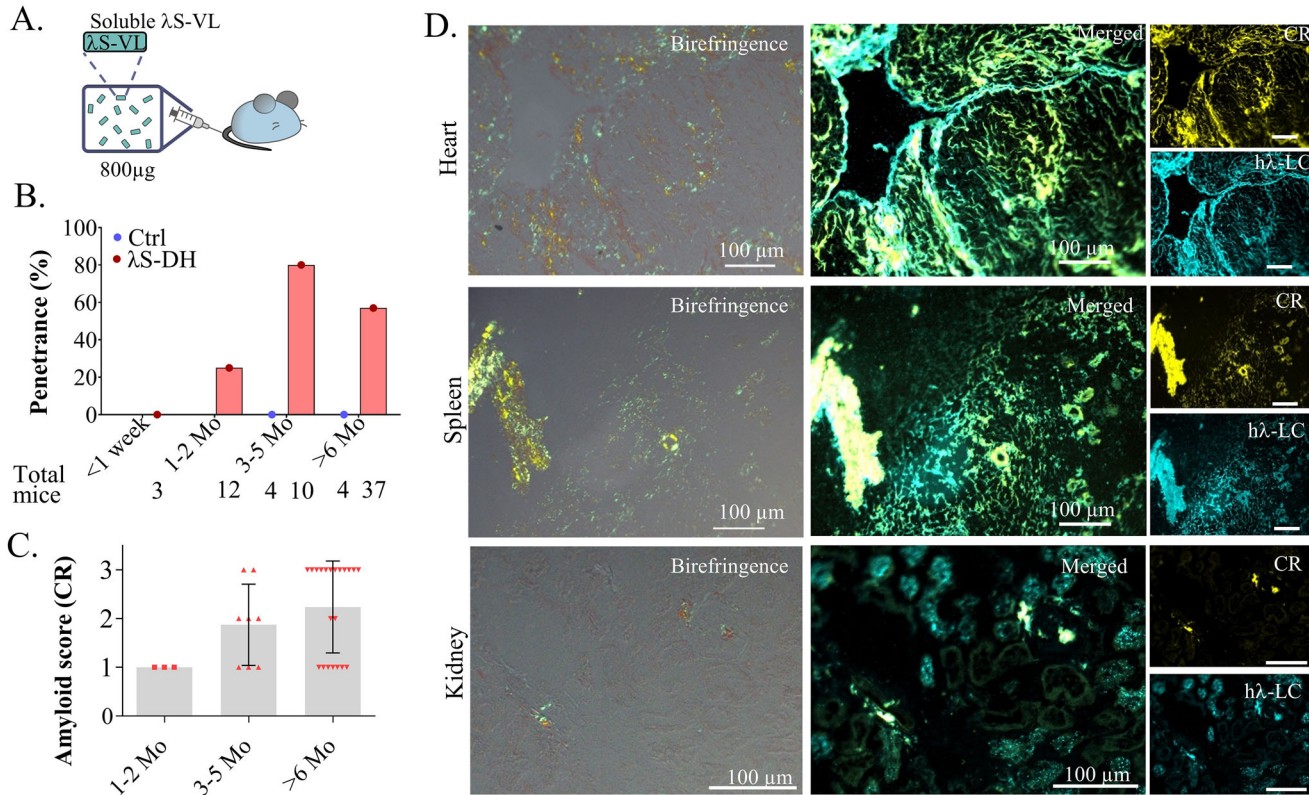

**Fig. 5 | Amyloid deposits induction in λS-DH mice with soluble λS-VL. A** Protocol of amyloidosis induction with soluble rλS-VL. **B** Organs from induced mice (λS-DH, red, and controls, blue) were analyzed histologically with a CR staining at different time points after the injection. AL-positive mice were determined by the presence of CR fluorescence in the heart, and the penetrance was calculated as the ratio of positive mice to the total number of mice analyzed at each time point (indicated below the graph). **C** The amyloidosis score (1, 2, and 3) was calculated for the

positive mice in the cardiac tissue ($n = 3$ at 1–2 months (Mo), $n = 8$ at 3–5 months, and $n = 21$ at >6 months). Representation of the single values with mean (gray bar) ± SD. **D** Histological characterization of mice deposits with CR staining revealed birefringence (left) and fluorescence (right) of the CR and colocalization with the anti-hλ-LCs antibody in the heart, the spleen, and the kidneys. Representative images of a score 3 mouse.

fragmentation pattern of the LCs between patients and our mouse model also suggest a conserved mechanism of amyloid fibril formation.

## AL fibrils deposition induces early cardiac dysfunction in mice

We next sought to determine if AL deposits in the heart of λS-DH mice could lead to cardiac dysfunction. The more sensitive and specific biomarker of cardiac amyloidosis in humans is NT-proBNP. Unfortunately, at the time most experiments of the present study were done, we could not find a reliable NT-proBNP assay kit for mice. Those tested showed high assay variability in control groups and inconsistencies between kits, ranging from pg to ng on the same samples. Recently, we eventually found a new assay, which gave more reliable results and we retrospectively measured the concentration of NT-proBNP in the serum of mice with confirmed cardiac amyloidosis (AL, $n = 13$, mean score = 2. 77, range = 2–3) (Fig. S6A) and compared it to the levels in DH mice (DH, $n = 14$) and AL negative λS-DH mice (LS, $n = 13$). We observed a significant increase between the AL and LS mice (mean 1832pg/ml ±418 vs 306 pg/ml ±42 respectively, $p = 0.0001$), and between the AL and the DH mice ($264 \pm 34.9$, $p < 0.0001$) albeit with a high heterogeneity in the AL group (Fig. 7A). No difference was observed between the LS and DH mice ($p = 0.6411$). We also analyzed two cohorts of 4 λS-DH mice 6 months after VL-seeding induction (14–20 months old at sacrifice) and 8 age-matched controls (DH) by high-resolution echocardiography. Thereafter, mice were sacrificed for histological studies. Upon the AL mice, two were excluded from the analysis: one did not present AL deposits at sacrifice. The other, with confirmed AL deposits, showed evidence of an intracardiac thrombus

in the left atrium (Fig. S6B) that could be related to the cardiac amyloidosis as commonly observed in patients[44] but precluded its use to study cardiac parameters. Similarly, in the control group, one died for an undetermined reason before the end of the experiment and another one presented with a significant tumor mass in the liver and a right kidney atrophy. Both were excluded from the analysis. As expected from previous experiments, we observed high heterogeneity in the amyloid burden of AL-positive mice (mean score 2.5, range = 1–3) (Fig S6C). We analyzed classical parameters used in humans to characterize cardiac AL amyloidosis[45]. Echocardiographic parameters were consistent with ventricular hypertrophy with an increased diastolic posterior wall thickness (LVPWd) in AL hearts (1.08 mm ± 0.112 vs. $0.790 \pm 0.030$ in controls, $p = 0.0087$) (Fig. 7B) and a tendency that did not reach significance for global ventricular hypertrophy (corrected LV mass, $p = 0.0649$) (Fig. S6E, F). We also detected an early diastolic dysfunction in AL mice, including higher left ventricular (LV) filling pressures (E/E′) ($43.3 \pm 5.26$ vs $26.1 \pm 1.48$ in controls, $p = 0.0087$) (Fig. 7C) and lower early diastolic strain rate (SRe) ($4.658 \pm 0.443$ vs $7.207 \pm 0.897$, $p = 0.0152$) (Fig. 7D), suggesting a reduction in passive ventricular relaxation associated with a preserved ventricular ejection fraction ($58.6\% \pm 5.56$ vs. $56.8\% \pm 1.72$, $p = 0.8182$) (Fig. S6G). Of note, ventricular ejection fraction values were slightly below 60% in both groups which is consistent with the progressive age-related decline observed in mice older than 12 months and not indicative of heart failure with systolic dyfunction[46]. Differences in the global longitudinal strain (GLS) were not significant between groups ($p = 0.7879$) (Fig. S6H) but a reduced ventricular filling capacity was observed by the increase in left atrial (LA) volume and a decrease in LA ejection

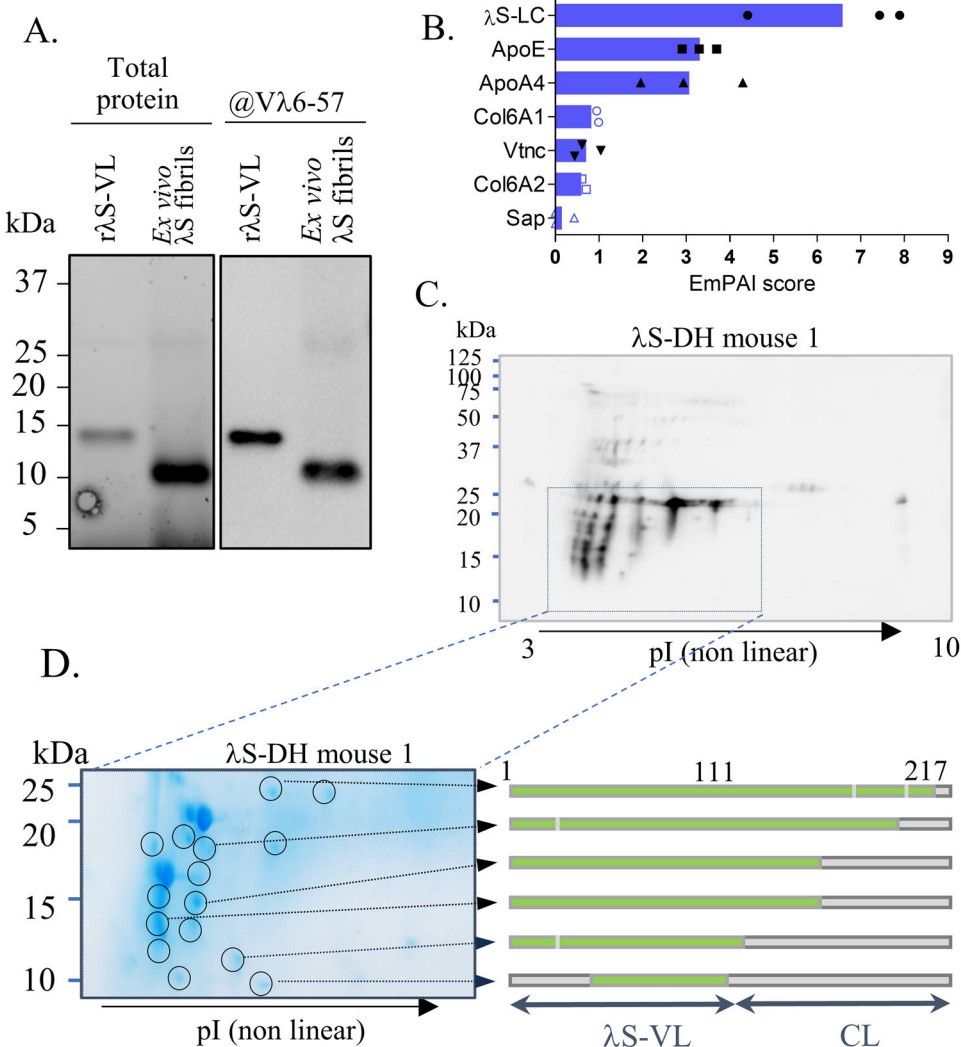

**Fig. 6 | Molecular characteristics of the amyloid deposits in λS-DH mice.**
**A** Analysis of the purified λS-DH fibrils by SDS-Page and Western Blot. Total protein was revealed by the Stain-Free technology (Biorad). Blotting was performed with an anti-human Vλ6-57 antibody. Images are representative examples from *n* = 3 independent experiments. **B** Mass spectrometry analysis of ex vivo purified fibrils obtained from 3 extractions (*n* = 5 hearts). The protein content of samples was given by the EmPAI scores (single values with mean). Only the amyloid signature proteins are shown. ApoE Apolipoprotein E, ApoA4 Apolipoprotein A4, Vtnc Vitronectin, Sap Serum amyloid P component, Col6A1 Collagen type VI alpha 1 chain, Col6A2 Collagen type VI alpha 2 chain (**C**) Analysis of deposited light chains in an amyloid-positive mouse by 2D-PAGE and 2D western blotting using anti-hλ LCs (*n* = 3). The boxed region matches the corresponding region of the image shown in (**D**). **D** Coomassie-stained gel from the boxed region in (**C**) (left). The spots whose position matches the immunoreactive spots in the western blot shown in (**C**) (circled) were excised, trypsin-digested, and analyzed by LC-MS/MS. The coverage of the monoclonal light chain sequence in each spot, from the tryptic peptides identified by LC-MS/MS, is represented in green on the right.

fraction (Fig. S6I). All these data are mainly indicators of heart failure with preserved ejection fraction.

## Transcriptomic analyses define extracellular molecular events but not cellular toxicity as a feature of amyloid fibrils toxicity in mice

To further decipher molecular mechanisms leading to cardiac dysfunction, we performed a bulk RNA sequencing on heart apical tissue from the AL positive λS-DH mice (*n* = 9, AL1-9, mean amyloid score 2.33, range = 1–3) (Fig. S6J) and age-matched DH mice as controls (*n* = 7, Ctrl1-7). We also analyzed non–induced amyloid negative cardiac tissue from λS-DH (*n* = 4, LS1-4) to identify any direct toxicity of the circulating amyloidogenic LC. Using Principal Component Analysis (PCA) and hierarchical clustering of the samples showed a clear separation between AL mice and the other groups (LS and Ctrl) (Figs. 7E and S6K). However, the two AL negative groups showed no clear distinction, whatever the chosen comparison (AL vs Ctrl or AL vs

LS). The transcriptional changes (DEG, log2FC > 0.58, and FDR < 0.05) between these three groups revealed that 1135 and 1766 genes were differentially expressed between AL and Ctrl and between AL and LS samples, respectively. Comparison between LS and Ctrl confirmed the PCA and hierarchical clustering with only 3 significantly deregulated genes with irrelevant functions (Table S1). Since LS and Ctrl groups were not distinguishable, we decided to merge them into a unique group of control samples to strengthen our statistical analysis. This new analysis (log2FC > 1 and FDR < 0.05) resulted in 2034 genes that were differentially expressed (1093 upregulated and 941 downregulated) (Fig. S6L). For gene set enrichment analysis (GSEA)[47], we used the WEB-based Gene Set Analysis Toolkit (WebGestalt)[48] to study the deregulated pathways (Reactome database)[49] and gene ontology Molecular Function, Cellular Component and Biological Process signatures of the amyloidogenic cardiac tissue. Despite the number of DEG, only 9 pathways, all positively enriched, were significant (FDR < 0.05) (Fig. S6M), and many of them were closely related so we used the

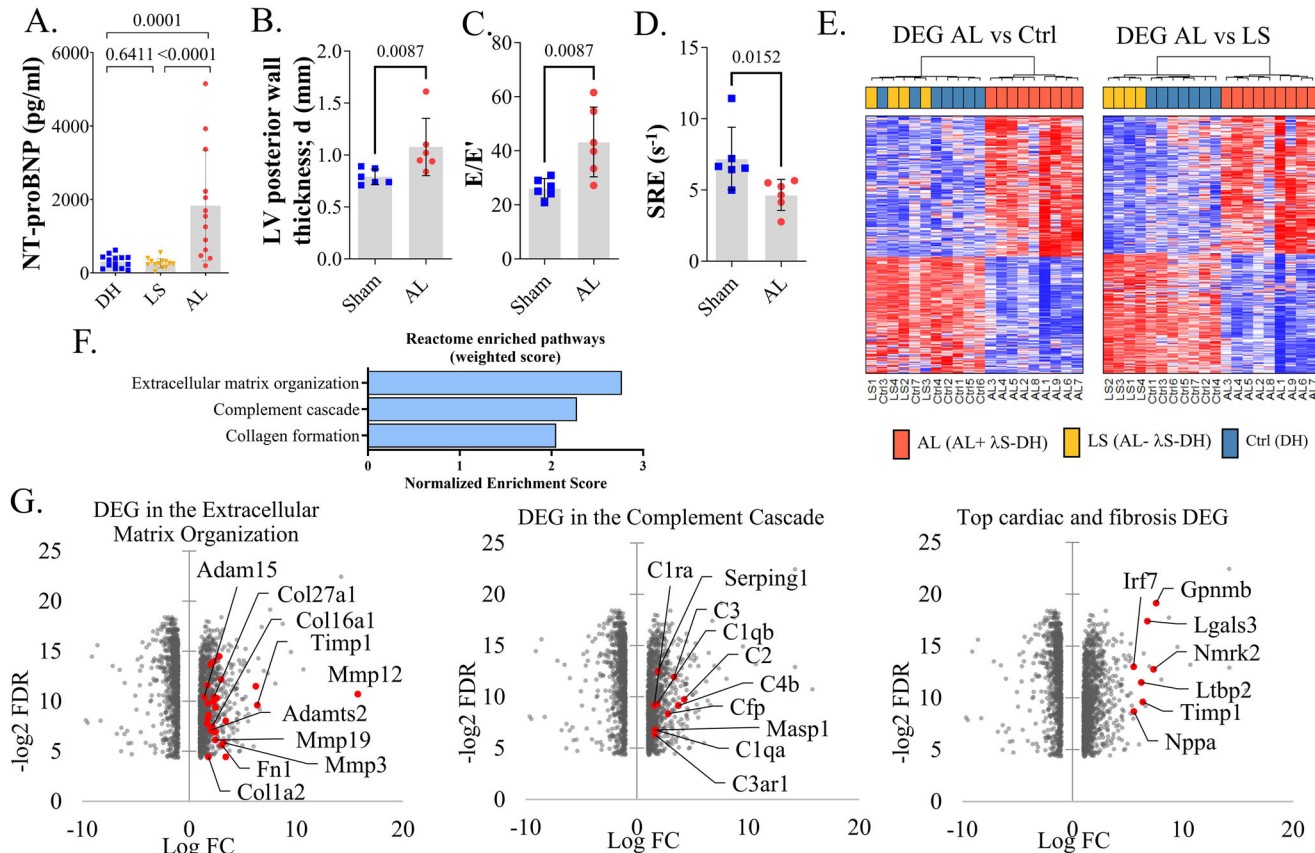

**Fig. 7 | Analysis of the cardiac function related to amyloid deposition in λS-DH mice. A** Plasma NT-proBNP was dosed in score 2-3 amyloid-positive λS-DH (AL, $n = 13$, red), λS-DH mice without amyloid deposition (LS, $n = 13$, yellow) and DH mice ($n = 14$, blue). **B** Cardiac measurement of the diastolic (d) left ventricular (LV) posterior wall by Ultrasound analysis in age-matched DH control mice (Sham, $n = 6$) and λS-DH with a cardiac amyloid deposition (AL, $n = 6$) (left). **C** Cardiac filling pressure, given by the E/E′ ratio analyzed on the same mice as in (**B**). **D** Cardiac early diastolic strain rate (SRE) analyzed as in (**B**). **E** Heatmaps showing the differentially expressed genes (DEG, log2 fold change |FC| ≥ 0.58 and False Discovery Rate FDR < 0.05) in λS-DH mice with cardiac deposits (AL, $n = 9$) compared to DH mice

(Ctrl, $n = 7$, left) and compared to λS-DH mice without deposits (LS, $n = 4$, right). Mice were clustered according to the DE profiles. Red corresponds to over-expressed genes and blue to downregulated genes. **F** Weighted scoring representation of the enriched Reactome pathways (associated to the 2034 DEG (log2 |FC | ≥ 1 FDR < 0.05)) when comparing AL vs Ctrl + LS ($n = 11$). **G** Overexpressed genes associated to the extracellular matrix organization, complement cascade or cardiac dysfunction, and fibrotic processes pathways according to their FC (log2) and their FDR (−log2). Histograms show individual values with mean ± SD. Comparisons in (**A**, **D**) were performed by Mann-Whitney two-sided test. Exact *P* values are indicated.

weighted set cover option to avoid redundancy (Fig. 7F). The most enriched pathways were related the extracellular matrix (ECM) organization (5 out of 9 statistically enriched pathways), with 41 differentially expressed genes encoding for endoproteases (MMPs, ADAMs, and extracellular cathepsins) and their inhibitors (TIMPs), as well as ECM-related proteins, such as collagens, fibronectin, laminin and adhesion molecules (Figs. 7G and S6N). Other pathways comprised complement cascade (4 out of 9 enriched pathways) and Collagene formation (Figs. 7F and S6M). Gene Ontology processes associated with the DEG corroborated these findings (Fig. S6O), and immune cell populations were principally enriched in the cell-type landmarks analyzed by the Mouse Cell Atlas (Fig. S6P). Strikingly, we did not identify any obvious molecular signatures of cellular toxicity in the amyloid cardiac tissue. However, we detected several known markers of cardiac dysfunction and cardiac fibrosis among the top 20 overexpressed genes (Fig. 7G) including *Nppa, Lgals3, Gpnmb, Timp1, Nmrk2, Ltbp2, Irf7*[50]. Most of these markers were also overexpressed in AL-positive hearts compared to each control group (LS or Ctrl) independently (Table S2). *Nppb* overexpression did not reach significance (Log2FC = 2.18 and FDR = 0.054), likely due to a high variability between mice. Collectively, these findings indicate that the soluble amyloidogenic λS-LC does not seem to exert any obvious toxic effect on the cardiac tissue of our transgenic mice. Nevertheless, amyloid

deposition demonstrated clear effects on the extracellular compartment of cardiac tissue, consistent with extracellular matrix remodeling, early fibrotic processes, and immune cell infiltration.

## Discussion

We herein describe the first transgenic mouse model of systemic AL amyloidosis with dominant cardiac involvement. Thanks to a strategy coupling an orthotropic production of the human LC by plasma cells and the removal of endogenous HC to force the production of fLCs, these mice continuously produce a level of circulating fLC that is above the median level observed in AL patients[51], resolving one of the main limitations of previous models[22–24]. Upon induction, mice accumulated AL amyloid fibrils in the heart, spleen, and, to a lower extent, in other visceral organs. These fibrils have typical pathological features of λ LC AL amyloid deposits with most of the classical amyloid signature proteins (ApoE, ApoA-IV, Vitronectin) except SAP which was barely detectable by MS and not detected by IF. This contrasts with previous models of ATTR amyloidosis[37,52] although it may be explained by the low level of circulating SAP in our mice and the weak avidity of mouse SAP to amyloid deposits[53]. To achieve amyloid deposition in λS-DH mice, we used two different induction methods. One is the classical seeding strategy with the injection of preformed fibrils that accelerates amyloid formation by bypassing the long early stages of nucleation.

We injected in vitro fibrils exclusively composed of the recombinant VL part of λS-LC because tissue from λS patients to make ex vivo fibrils was unavailable and because we failed to obtain any fibrils from the recombinant full-length λS-LC. For the second protocol, we injected the soluble VL alone assuming that its intrinsic amyloidoigenicity would allow its aggregation in vivo. Both protocols led to amyloid deposits in mice but with different timing and incomplete penetrance. Whereas the fibril injections were able to trigger a rapid burst of amyloidosis, detectable as soon as 48 h after induction, the VL injections led to a slower, more physiological development of the disease. Due to the rapid renal clearance of low molecular weight proteins, we assume that a few oligomers of the VL are quickly formed in the circulation or tissues where they remain progressively elongated by full-length LC and spread to other organs through secondary nucleation and fragmentation[54]. Although we cannot formally exclude that the injected material may have contained a tiny amount of pre-formed amyloid seeds below the limit of detection by HPLC, this result means that once formed, these oligomers likely persist for a long period of time in tissues. This is, to our knowledge, the first in vivo experiment supporting the enhanced potential of the VL to initiate the subsequent formation of mature amyloid fibrils elongated by full-length LC. We conveniently used the strict VL fragment based on our experiments, the literature about in vitro fertilization, and the cryo-EM structures of AL fibrils but we assume that the pathophysiological process in humans is likely more complex than a simple, precise, release of the sole VL domain. Accordingly, Lavatelli et al. recently showed in vitro that a partial degradation of the constant domain, as frequently observed in AL deposits, is sufficient to unleash the amyloidogenic propensities of an otherwise stable full-length LC[8]. Consequently, based on our findings, it is tempting to hypothesize that similar partial proteolysis of fLCs, releasing fragments containing destabilized VL that may rapidly form the first protease-resistant nuclei, occurs in plasma or extracellular matrix of organs and may initiate the elongation of amyloid fibrils with the circulating monoclonal full-length LC. However, in the absence of a formal experimental demonstration, we cannot exclude that in patients or in the few mice that developed spontaneous amyloidosis, the VL contained in the intact fLC was able to progressively acquire its amyloidogenic conformation, exposing the CL at the periphery of the fibrils, leading to their progressive degradation by activated resident proteases. In any case, the characterization of mature amyloid fibrils in tissues or extracted from hearts of positive λS-DH mice opens fundamental perspectives on the mechanisms of amyloid formation in vivo. First, these analyzes showed that the insoluble deposits do not exclusively contain the VL used for seeding, but they incontrovertibly indicate that the endogenous full-length LC is also incorporated. Consequently, whether a critical concentration of VL-containing fragments is needed for priming the process, the full-length LC subsequently or concomitantly participates in amyloid progression. Although the events that generate the complex population of deposited LC fragments are not known, these proteolytic events are virtually identical both in the different mice and also in human LC deposits. As in human amyloid, major fragments in mice contain the complete VL and progressively shorter stretches of the CL, suggesting that the VL constitutes the core of the murine AL amyloid fibrils, as previously shown in human[9,55]. The pattern also suggests that the proteolytic remodeling mechanisms associated with LC aggregation and/or aggregate metabolism are analogous in mice and humans. With this in mind, it would be useful to identify the potential molecular players of this proteolytic event, their location, and activation. Interestingly, transcriptomic analyzes of amyloid cardiac tissue show significant overexpression of several metalloproteinases and extracellular cathepsins that could account for the LC fragmentation. It remains to be determined whether these proteolytic events participate in or accelerate the elongation of amyloid fibrils or, on the contrary, whether they are involved in their decay.

With very rare exceptions, mice do not spontaneously develop AL amyloidosis, as previously shown in other systemic amyloidosis models[37]. There are several hypotheses to explain this observation, including a faster turnover rate of proteins or a better control of proteostasis in mice[26,27]. The low amyloidogenicity of the λS-VL protein compared to other published IGLV6-57 domains[35,56] and the absence of associated serum amyloid P component to the fibrils, known to protect fibrils from degradation[42], could also explain this observation. However, in humans, the production of fLC precedes several years of the occurrence of symptomatic AL amyloidosis[39], and in other systemic amyloidosis, especially hereditary ones, the onset of the disease usually after several decades. Accordingly, in λS-DH mice, amyloidosis development takes several months even when accelerated with a large quantity of amyloidogenic VL. Then, we cannot exclude that even with the pathogenic level of fLC observed in humans, mice simply do not live long enough to let the time to allow overt amyloid deposition to develop. In any case, the rare spontaneous AL amyloidosis observed during the course of this study demonstrates that mice are not intrinsically resistant to AL amyloidosis. Since an increase in fLC level precedes the onset of clinical symptoms, reaching higher levels of fLCs in future transgenic mice could help accelerate the natural onset of the disease.

In contrast, the organ tropism seems to be relatively distinct in the mice compared to the patient from whom the transgenic LC was extracted. Amyloidosis in λS-DH mice develops mainly in the heart and associated vessels. At odds with the patient who was first diagnosed with renal involvement, the deposits in kidneys appear late and remain restricted to a few isolated glomeruli. One could speculate either that the tropism for a kidney of the LC or the injected seeds was low because of missing receptors that could bind and retain them into glomeruli or the mouse kidneys express factors that protect them from amyloid formation (efficient extracellular proteostasis). But we cannot exclude technical issues like the i.v. route for seeds and VL injections or the higher level of fLC in the mice compared to human could also, for unknown reasons, favor their early deposition in the heart at the expense of the kidney or other organs. In any case, understanding this discrepancy in organ tropism between humans and mice is likely to provide clues about the mechanisms leading to amyloidosis in a preferential organ.

In addition to the formation of amyloid deposits, this model also develops early symptoms of cardiac dysfunction. We found a significant increase of NT-proBNP, the most reliable plasmatic biomarker for AL amyloidosis, in AL-positive mice compared to non-induced λS-DH or DH control mice. Induced λS-DH developed also early signs of diastolic dysfunction with preserved ejection fraction. Although incomplete, this dysfunction shows that beyond AL deposits, λS-DH can also be a relevant model to study the molecular events leading to cardiac toxicity. Accordingly, transcriptomic analyzes showed significant overexpression of genes classically associated with cardiac dysfunction, especially those involved in fibrosis and extracellular matrix remodeling. Although we did not detect significant infiltration of immune cells in the cardiac tissue, transcriptomic data indicate that myeloid-derived cells, including macrophages, monocytes, or dendritic cells, together with cardiac fibroblasts may account for this profibrotic phenotype. Deciphering the location and function of these cells will likely give clues about their role during AL amyloidosis development. For this purpose, spatial transcriptomic and/or proteomic on mouse tissues with different degrees of amyloid deposition would likely be of great value, especially to better understand changes in cell populations and their activity in the vicinity of amyloid deposits.

There are two major discrepancies with previous in vitro or non-mammalian models. First, based on transcriptomic analysis and histology, there is no clear indication of cardiomyocytes toxicity, induced by the fibrils. We may hypothesize that mice only develop the early stages of the disease, as seen by the discreet cardiac hypertrophy and

amyloid loads below those observed in human cardiac biopsies[57]. Longer exposure and further accumulation of deposits would ultimately lead to myocyte damage. The second discrepancy concerns the apparent absence of cardiac toxicity caused by the soluble amyloidogenic LC. The toxicity of amyloidogenic fLCs has been extensively studied on cardiac cells (cardiomyocytes, cardiac fibroblasts, and mesenchymal cells)[14,16,18] in vitro, as well as in lower animal models, such as *C. elegans* and zebrafish[15,17]. Among the observed effects, mitochondrial dysfunction, ROS production, and cell death were directly associated with soluble fLCs. In the present mouse model, the production of the LC by plasma cells continuously exposes cardiac cells to a high level of an amyloidogenic LC. However, mice with no amyloid deposits seem healthy, with no increase of NT-proBNP and a transcriptional landscape of cardiac tissue almost identical to control mice. There are several hypotheses that may be proposed to explain this difference. First, the patient from whom the *λS-LC* gene was extracted was first diagnosed with renal involvement so one could hypothesize that it is not a cardiotoxic LC per se. However, even in the absence of direct evidence of cardiac deposits in the patient, his clinical symptoms were characteristic of AL amyloidosis-induced cardiac dysfunction. Then obviously, mice are not men, and differences in protein turnover, extracellular protein chaperoning, cellular resistance to LC-induced stress, or absence of LC binding to cell surface could all account for the undetectable LC toxicity in our mouse model[16,18,26,27,58]. On the other hand, the direct toxicity of amyloid LCs on cultured cells, ex vivo organs, or in lower animals could also be challenged since these systems do not reproduce the in vivo extracellular proteostasis, including circulating chaperones and proteases, which could protect cells from unstable or unfolded dangerous LCs in the mice or in humans[59]. In addition, the immune system, extracellular matrix environment, or cell type diversity, all potentially involved both in the pathogenicity and also the protection of organs from toxic LCs, cannot be reproduced in vitro or in lower organisms. Finally, even in the absence of direct toxicity, λS-DH mice develop an early cardiac dysfunction closely resembling the human disease. This dysfunction relies on the presence and accumulation of amyloid fibrils that progressively replace and remodel the extracellular space of cardiac muscle, likely causing mechanical constraints as a first step. Then, further accumulation of amyloid fibrils, reaching a critical threshold, would likely lead to cellular toxicity and more severe cardiac dysfunction. The quick recovery of the cardiac function in patients upon the reduction of circulating LC remains to be understood but could rely on the arrest of further accumulation of new toxic fibrils.

In addition to its contribution to the pathophysiological mechanisms of AL amyloidosis, our model also fills a crucial gap in the validation of new therapeutic approaches. Having shown that amyloid deposits are very similar to those observed in humans, it is now possible to use this model to test new therapeutic approaches designed to eliminate amyloid fibrils in tissues or to stabilize/inhibit the circulating amyloid precursor. The absence of Ig heavy chain, and therefore of a proper humoral immune response, is of particular interest in this respect, as it allows the repeated injection of humanized molecules without any risk of immune rejection. The main limitation to the use of this model for pre-clinical purposes remains the incomplete penetrance of the disease but modifications of the induction protocol by repeated seeding or ex vivo fibril injection are currently being evaluated.

Overall, the transgenic λS-DH mouse is a representative model of cardiac AL amyloidosis and a useful resource for studying the formation, composition, and metabolism of human AL deposits. We confirm the crucial role of the variable domain in initiating amyloid deposits and provide new insights into the toxicity of amyloid fibrils for the heart. This model offers a new avenue for research on AL amyloidosis and fills an important gap for the development of new therapies. Other models using the same strategy but producing LCs from patients with

different organ involvement need to be developed to better represent and understand the diversity of this disease.

## Methods

### Clinical presentation of λS patient

Patient FLC levels at diagnosis were 128 mg/L for λ and 18 mg/L for κ-LC, associated with an IgAλ monoclonal component in serum, and 6.5% of medullar plasma cells (PC) infiltration. Clinical manifestations at diagnosis included chronic kidney disease (serum creatinine level 191 μmol/l, hypoproteinemia 56 g/l, heavy proteinuria 5.8 g/24 h) and typical features of amyloid cardiomyopathy (diffuse microvoltage on ECG, interventricular septum thickness of 14 mm, diastolic dysfunction, preserved left ventricular ejection fraction at 60%). Serum NT-proBNP was initially 13,173 pg/mL. A kidney biopsy confirmed the presence of AL amyloid deposits composed of a λ-LC (Fig. S1A). Written informed consent was obtained from all study participants and this study has received approval for the retention and treatment of human biological samples from the Comité de Protection des Personnes (CPP DC-2008-111).

### λS LC gene extraction and sequencing

The human monoclonal LC genes used in the study were obtained from the patient's bone marrow (BM) aspirates. Total RNA was extracted using Trizol® reagent (Invitrogen) and reverse transcription was carried out with High Capacity cDNA Reverse Transcription Kit (Applied Biosystems). For λS-LC, cDNA was amplified using a 5′ consensus primer in the leader sequences (5′ ATGGCCTGGDYY-VYDCTVYTYCT 3′) and a 3′ consensus primer complementary to the different λ constant regions (5′ CTCCCGGGTAGAAGTCACT 3′). For κR-LC, cDNA was amplified using a 5′ consensus primer in the leader sequences (ATGAGGSTCCCYGCTCAGCTC) and 3′ reverse primer in the κ constant region (GCGGGAAGATGAAGACAGAT). PCR products were purified and cloned using TOPO TA cloning Kit (Invitrogen) and Sanger sequencing was performed using the M13 reverse primer and the Big Dye Terminator Cycle sequencing kit (Applied Biosystems). The AL *λS-LC* gene was extracted from a patient with biopsy-proven AL amyloidosis (λS-PT) and derived from the *IGLV6-57* germline gene (95.3% homologous) rearranged on an *IGLJ3* junction segment (92.11% homologous) and a constant *IGLC3* domain. The κR-LC gene (*IGKV2-28/IGKJ2*) used to generate the κR-DH model was extracted from a patient with multiple myeloma and cast nephropathy, with no amyloidosis in the kidney biopsy and no other clinical argument to suspect amyloidosis.

### Transgenic mice models generation

To generate the transgenic mice models (λS-DH and κR-DH) two similar strategies were used[30,31]. For both models, a 2.2 kb fragment containing the five Jκ mouse segments was replaced with a cassette containing the mouse pVH promoter, the rearranged LCs genes as described below, and a floxed Neomycin resistance cassette. For λS-DH, the complete cDNA preceded by a β-globin/IgG chimeric intron (from pCi vector, Promega) and followed by the bGH poly A signal was introduced in place of the mouse *Jκ* segments in the κ locus thus generating a fully human LC. For κR-DH, the cDNA coding for the variable/junction domains (VJ) was introduced, generating in the mouse a chimeric human/mouse LC composed of the human VJ domain and the mouse C domain. These constructs were transfected into E14 129 ES cells, selected with Neomycin (Fisher Bioreagent) and Ganciclovir (Sigma–Aldrich), and injected into C57BL/6 blastocysts. Positive F1 mice were crossed with a CMV-Cre deleter strain (B6.C-Tg(CMV-cre)1Cgn/J, The Jackson Laboratory) to remove the Neomycin cassette and then crossed with DH-LMP2A mice, kindly provided by S. Casola (IFOM, Milan, Italy)[33], to generate the double homozygous strains. The transgenic strategy for λS-DH is shown in Fig. 1A. Mice were maintained in pathogen-free, constant temperature (20–24 °C) and

humidity (50% ± 10) conditions, following the French regulations, with food (SAFE® 150 pellets from SAFE diets, France) and water *ad libitum* and 12 h dark, 12 h light cycles. Monitoring of the animals was daily performed by researchers and animal care staff. All procedures used for the characterization of $\lambda$S-DH mice (flow cytometry, biochemical parameters, fLC dosage, western blot) are detailed below. Experiments were conducted in both male and female adult transgenic mice (>3 months old). Mice were sacrificed using increasing doses of $CO_2$ gas at specific time points. All experimental procedures have been approved by our institutional (university of Limoges) review board for animal experimentation (Comité régional d'éthique sur l'expérimentation animale du Limousin−CREEAL) and of the French Ministry of Research (APAFIS #7655-2016112211028184).

### Histological studies

Organ samples for immunofluorescence (IF) studies were OCT-embedded and snap-frozen in isopentane using a Snap Frost 2 (Excilone) immediately after sacrifice. IF was performed on 9 µm cryosections fixed with cold acetone, blocked in PBS with 3% BSA, and then stained with appropriate antibodies and/or CR (Table S3). For Congo Red (CR) staining, 2.87 mM CR alkaline solution was freshly prepared by adding 200 µl of 1% NaOH to 20 ml of 250 mM NaCl and 80% Ethanol CR working solution followed by a 5-min staining before washing with PBS. Slides were observed on a NiE microscope (Nikon). For electron microscopy (EM), tissues were fixed in 4% glutaraldehyde in 0,1 M phosphate buffer (pH 7, 2) at 4 °C, embedded in resin, and ultrathin sections were visualized on a JEOL JEM-1010 electron microscope (JEOL Ltd). For immunogold studies, sections were additionally incubated with an anti-human lambda gold-conjugated antibody. For histochemical staining, tissues were fixed in 4% paraformaldehyde, paraffin-embedded and the slides were stained either with Hematoxylin-Eosin and Congo Red or Masson's Trichrome. To evaluate the amyloid burden in hearts, a score was established based on the CR fluorescence as follows: score low (=1) corresponds to one or more focal deposits in the myocardium and/or blood vessels; mid (=2) corresponds to some diffuse deposits within the myocardium and along most vascular walls; and high (=3) corresponds to diffuse deposits throughout the myocardium and along all vascular walls.

### Recombinant LC and VL production and purification

To produce the full-length $\lambda$S-LC, the cDNA obtained from the patient was amplified and cloned in a modified pCpGfree plasmid (Invivogen, San Diego, USA) containing the neomycin resistance gene. Series of transfections in the murine hybridoma/myeloma cell line Sp2/0-Ag14 (ATCC® CRL-1581™) were performed using Cell line Nucleofactor kit V (Amaxa/Lonza, Basel, Switzerland) and a Nucleofactor II device (Amaxa/Lonza, Basel, Switzerland). Positive clones were then selected using neomycin (1 mg/mL, Fisher Bioreagent, Pittsburg, USA). Best producing SP2/0 $\lambda$S-LC clone was cultured in a mini-bioreactor system (CELLine CL 350, Integra Biosciences), and the secreted fLC was collected according to manufacturer's instructions. All collected samples were pooled and fLCs were purified by affinity chromatography using fLC-sepharose resins (a kind gift from Binding Site, Birmingham, UK) or Lambda-FabSelect column (Cytiva). Protein purity was assessed by SDS-PAGE and Coomassie Blue staining and the resulting purified protein was concentrated (3 kDa Amicon® Ultra filter, Merck, France) and quantified as described above. Recombinant human $\lambda$S-VL (r$\lambda$S-VL) was periplasmatically expressed in *E.coli* BL21 competent cells based on the pET12a vector (Novagene) containing an optimal peptide for periplasmic expression and the cDNA coding the VJ region of $\lambda$S-LC. Signal peptide cleavage left two extra amino acids in N-term (ST) compared to the human VJ sequence (Fig. S2A). After overnight growth, the proteins were extracted with a cold osmotic shock and purified on a POROS™ XQ anion exchange resin (Thermo Fisher Scientific) followed by a Ceramic Hydroxyapatite resin on an AKTA-FPLC system (GE Healthcare). Protein

purity was assessed by SDS-PAGE and Coomassie Blue staining and the concentration was determined by BCA assay (Thermo Fisher Scientific). Samples were stored in 10 mM Hepes, 100 mM NaCl, and pH 7.6. Absence of aggregated $\lambda$S-VL was monitored with a native SEC-UV/MS using an ACQUITY UPLC Protein BEH200 SEC column (4.6 × 150 mm, 1.7 mm particle size; Waters, Milford, MA, USA). An isocratic elution with water at 0.1 mL/min was used for chromatographic separation on a Nexera LC40 system (Shimadzu, Noisiel, France) equipped with UV detection at 280 nm. Sample injection amounts of 10 µg were used and data acquisition was controlled by Hystar (Bruker Daltonics).

### In vitro fibril formation

For aggregation assays, 30 µM of protein ($\lambda$S-VL or $\lambda$S-LC) was incubated in aggregation buffer (10 mM Hepes, 100 mM NaCl, 5 mM DTT, 0.8 µg/µl Heparin, pH 7.6) at 37 °C in an orbital shaker at 300 rpm for the appropriate time. For in vitro seeding assays, r$\lambda$S-VL seeds were prepared by sonication of $\lambda$S-VL fibrils 20 s at an amplitude of 90% using an Ultrasonic homogenizer (Bandelin). 0.6 µM seeds were mixed with $\lambda$S-VL or $\lambda$S-LC before starting the aggregation assay as mentioned above. Fibrilization kinetics was followed by ThT fluorescence (FLUOstar Omega, BMG Labtech), and confirmed by Atomic Force Microscopy (AFM) or Transmission Electron Microscopy (TEM). For TEM analysis, 10 µl of fibril sample was deposited on a formvar/carbon grid of 200 copper mesh (Agar Scientific), and excess liquid was removed. Negative staining was performed with Uranyless 30% ethanol (Em-Grade), washed twice with H2O, and air-dried. Grids were analyzed on a JEOL JEM-1400 Flash Transmission Electron Microscope (JEOL Ltd). For AFM analysis, 10 µl of a fibril sample was applied to a freshly cleaved mica surface (Plano GmbH). After 3 min incubation, the mica was washed with water and dried under airflow. The sample was scanned using the tapping mode on the AFM 5500LS (Agilent). Fibrils were centrifugated, resuspended in sterile water, and kept at 4 °C or −80 °C for short-term or long-term storage, respectively.

### Induction of amyloidosis in mice

Three to 12-month-old $\lambda$S-DH mice received one i.v. injection of 200 µl of 2 mg/ml r$\lambda$S-VL seeds (sonicated fibrils) or 4 mg/ml soluble r$\lambda$S-VL diluted in PBS. Mice were euthanized at different time points, from 48 h to 12 months after the injection. As controls, $\kappa$R-DH (producing a human monoclonal fLC), DH-LMP2A (producing mouse polyclonal fLCs), and C57/B6 mice were also injected with the same amount of fibrils or soluble proteins and euthanized at different time points for histological analysis.

### Ex vivo amyloid fibrils purification and characterization by mass spectrometry

Amyloid fibrils from unfixed mice heart tissue were either purified or enriched according to previously established protocols with minor modifications[43,60]. Briefly, for purification, frozen hearts were cut into 1 mm pieces, grinded on a 20 µm strainer, and homogenized on TC buffer (20 mM Tris, 138 mM NaCl, 2 mM CaCl$_2$, pH 8.0) at 4 °C with a Kontes pellet pestle homogenizer (DWK Life Sciences). After centrifugation at 3100 × g, 1 min at 4 °C, the pellet was washed with ice-cold TC buffer and centrifuged several times, before the incubation in a cocktail of collagenase from *C. histolyticum* (Sigma−Aldrich) and DNAse I (Roche) for 2 h at 37 °C and 750 rpm. After centrifugation (3100 × g for 30 min), the pellet was washed 5 times with ice-cold TE buffer (20 mM Tris, 140 mM NaCl, 10 mM EDTA, pH 8.0). The resulting pellet was resolubilized with the Kontes pellet pestle homogenizer in ice-cold water and centrifuged at 3100 × g 5' at 4 °C at least 8 times. Supernatant fractions, as well as the remaining pellet, were analyzed by SDS-PAGE. One of the more concentrated fractions was used for further MS analysis. Twenty micrograms of fibril proteins were incubated in 8 M urea (10-fold dilution) for 2 h, then reduced, alkylated, and digested with trypsin by filter-assisted sample preparation method.

250 ng of resulting peptides were analyzed on a nanoElute2 nano-HPLC system (Bruker Daltonics) that was coupled to a TimsTOF Pro 2 mass spectrometer equipped with CaptiveSpray source (Bruker Daltonics). Peptides were separated on a PepSep C18 column (25 cm × 150 μm,1.5 μm (Extreme) Bruker Daltonics) by a linear 30 min water-acetonitrile gradient from 2% (v/v) to 25% (v/v) of ACN. The timsTOF Pro 2 was operated in DDA-PASEF mode (1.1 s standard method) using Compass Hystar 6.2. raw data were analyzed by Proteoscape software using the mouse Swiss-Prot database augmented with the sequence of λS-LC. EmPAI scores were used for analysis and only proteins with at least 2 stripped peptides were considered.

## Amyloid fibrils enrichment and proteolytic pattern of LC

For fibril enrichment, specimens were minced with a scalpel, washed 3 times with 1 ml of PBS containing protein inhibitors (Complete, Roche, Basel), and then manually homogenized in 250 μl of Tris EDTA buffer (20 mM Tris, 140 mM NaCl, Complete protein inhibitors, pH8.0) with a potter pestle. The homogenate was centrifuged for 5 min at 3100 × g at 4 °C and the pellet was retained. This step was repeated 10 times overall to remove proteins soluble in the saline solution. The final pellet, enriched in amyloid fibrils and other insoluble proteins, was retained for solubilization and analysis. Two-dimensional polyacrylamide gel electrophoresis (2D-PAGE) and western blotting were performed as follows[12]. Briefly, pellets containing enriched fibrils were incubated for 2 h with 8 M urea, 4% CHAPS, and 0.1 M DTT, followed by protein quantification in supernatants using BCA assay (Thermo Fisher Scientific, Waltham, MA, USA). Fifteen micrograms of proteins were analyzed by SDS-PAGE on 4−20% acrylamide gradient gels (Biorad), under reducing conditions. For 2D-PAGE analysis, proteins (120 μg for Commasie staining; 50 μg for western blotting) were diluted in Destreak Solution (Cytiva, Merck) plus 0.02% v/v pI 3−10 ampholytes (Bio-Rad). IEF was performed using 11 cm strips, non−linear 3−10 pH gradient (Bio-Rad), followed by SDS-PAGE separation on 8−16% polyacrylamide gradient midi gels (Criterion TGX gels, Bio-Rad). IPG strips were rehydrated at 50 V for 12 h, followed by IEF in a Bio-Rad ProteanTM IEF cell. Proteins were reduced (DTT 0.1 M) and alkylated (IAA 0.15 M) between the first and second dimensions. Gels were stained with Coomassie blue (Pierce, Thermo Fisher Scientific, Waltham, MA, USA). For Western blotting, proteins were transferred onto a PVDF membrane (Bio-Rad) and probed with polyclonal rabbit anti-human λ LCs (Dako, Agilent, Santa Clara, CA, USA) used at a concentration of 1 μg/ml, followed by incubation with a horseradish-peroxidase conjugated swine anti-rabbit secondary antibody (Dako). Imaging was performed with an ImageQuant LAS 4000 apparatus (GE Healthcare).

## Analysis of protein spots using liquid chromatography-tandem mass spectrometry (LC-MS/MS)

Protein spot excision and in-gel digestion were performed as described in ref. 9. Peptides extracted from each spot were analyzed by tandem mass spectrometry using an Ultimate 3000 nanoLC system combined with Thermo Scientific™ Q-Exactive Plus Orbitrap mass spectrometer. Tryptic digests were first loaded on a Thermo Scientific Acclaim PepMap C18 cartridge (0.3 mm × 5 mm, 5 μm/100 Å) and then chromatographed on a Thermo Scientific Easy-Spray Acclaim PepMap C18 column (75 μm × 15 cm, 3 μm/100 Å packing). MS data were processed using the Mascot software (Matrix Science, London, UK) searching into the Swiss-Prot Mus Musculus database, augmented with the sequence of λS-LC.

## High-resolution ultrasound echocardiography

High-resolution data were acquired using the Vevo 3100 high-frequency ultrasound system (FUJIFILM VisualSonics, Canada), equipped with a 40-MHz center frequency MX550D probe, to evaluate left ventricular systolic and diastolic function. Mice were anesthetized with 2.5% isoflurane at 1 ml/min air, and their body temperature was maintained at

37 °C using a heating pad. The mice were secured on a stage for imaging, with the ventral thorax hair removed. Throughout the imaging process, the electrocardiogram and respiratory rate were monitored. We performed 2D echocardiography according to the American Physiological Society guidelines for cardiac measurements in mice[61]. The longitudinal strain was analyzed in the parasternal long-axis (PLAX) view to obtain left ventricular (LV) ejection fraction and LV longitudinal strain and strain rate at early diastole (SRe). A four-chamber view in B-mode was obtained to evaluate diastolic dysfunction using pulse wave (PW) Doppler to assess isovolumic relaxation time, E/A ratio, and E/E' ratio. A short axis view was used to assess anterior and posterior wall thickness in diastole and systole. LV mass corrected was calculated using this formula: $0.8424 \times [(LVID;d + LVPW;d + IVS;d)^3 − LVID;d^3]$. To assess left atrium anatomy and function, we used 4D ultrasound (US) imaging. The US probe was attached to a linear step motor scanning in the parasternal short-axis (PSAX) view from below the apex to above the aortic arch. Acquisition parameters were set to a gain of 48 dB, a 3D range of 12−15 mm, 3D step size of 0.127 mm, and a frame rate of 400 frames/s. Each image acquisition required 10−15 min to create 4D data throughout one representative cardiac cycle from base to apex. We evaluated the atrial volume at systole and diastole and calculated the left atrium (LA) ejection fraction using the formula: stroke volume/end-diastolic volume. These three measurements provide insights into the heart's filling and relaxation. Offline image analysis was conducted using Vevo Lab Software 5.8.1 (FUJIFILM VisualSonics). Full data are available upon request.

## Transcriptomic analysis

Total RNA was extracted from the apical part of the hearts of λS-DH and DH-LMP2A mice using the miRNeasy Mini Kit (Qiagen) on the TRIzol reagent and the purity of the samples was assessed on the Bioanalyzer RNA 6000 Nano (Agilent) before generating the libraries from 500 ng of total RNA using the Truseq Stranded mRNA Kit (Illumina), quantified with the Qubit dsDNAHS Assay Kit (Invitrogen). RNA-sequencing was performed on a NovaSeq6000 (Illumina) with a flow cell S4 and a 2 × 150 bp paired-end chemistry. Paired-end reads were processed through the bioinformatic pipeline nf-core/RNAseq v3.12[62] in reverse strandedness mode. This performed quality control, alignment to the GRCm38 mouse genome with STAR[63], and quantification by Salmon[64] with the corresponding genome annotation from the ENSEMBL release 111 database. The batch effect due to the two separate sequencing runs was taken into account through ComBat adjustment[65] and the differential analysis performed by EdgeR (Bioconductor)[66]. Genes were considered differentially expressed (DE) if they had a corrected $p$ value (FDR) ≤ 0.05 and an absolute fold-change ≥ 1.5. GSEA analysis of the Reactome, Gene Ontology (GO) Biological Process, Cellular Components, and Molecular Function enriched pathways, as well as cell type enrichment by the Mouse Cell Atlas, were performed on WEB-based GEne Set Analysis Toolkit (WebGestalt)[67] using the ranked list of DE genes.

## Thermal stability of the proteins rλS-VL or rλS-LC

The concentration of the proteins was adjusted at 100 μg/ml in Hepes buffer (10 mM Hepes, 100 mM NaCl, pH7.6) and the thermal stability was measured on a Tycho NT.6 (NanoTemper Technologies) between 35 and 95 °C. The thermal unfolding midpoints ($T_m$) were calculated using the first derivative of the fluorescence intensity ratio at 350 nm /330 nm, corresponding to a conformational shift of tryptophan residues.

## LC dosage

Serums were analyzed for the presence of FLC using Freelite™ (The Binding Site, Birmingham, UK) assay on BNII nephelometer (Siemens healthcare, Herlangen, Germany) to the appropriate dilution for each sample and according to according to the manufacturer's instructions.

## Biochemical parameters determination

Biochemical parameters were measured on overnight urine collection and plasma samples, obtained by retro-orbital puncture under anesthesia or by cardiac puncture after their euthanasia on heparin. Plasma concentrations of creatinine were measured on a Konelab 30 analyzer with a creatinine enzymatic test (ThermoFisher Scientific). Urine albumin concentrations were measured using an albumin mouse ELISA kit (Abcam), according to the manufacturer's recommendations. Mouse N-Terminal Pro-Brain Natriuretic Peptide (NT-proBNP) was estimated in plasma using a mouse NT-proBNP ELISA kit (Elabscience).

## Western blot

Proteins were separated by reducing or non-reducing SDS-PAGE on Mini-Protean TGX Stain-free gels (Biorad). Total protein were detected by the fluorescence of the Stain-Free technology. After the proteins were transferred onto polyvinylidene difluoride membranes (Millipore), membranes were blocked in 5% milk Tris-buffered saline (TBS), followed by incubation with desired antibodies in 3% milk TBS (Table S3), washed three times with TBS 0.1 % Tween and revealed by chemiluminescence (ECL, Pierce).

## Flow cytometry

Splenocytes were isolated by tissue homogenization and stained with the appropriate antibodies (Table S3) in PBS 1x supplemented with 2% fetal calf serum. Flow cytometry analysis was performed on a CytoFLEX (Beckman Coulter) and data were analyzed with Kaluza. Intracellular staining was performed using the Intraprep™ kit (Beckman Coulter).

## In silico study of the λS-VL sequence

The aggregation prediction was generated on Aggrescan4D (https://biocomp.chem.uw.edu.pl/a4d/)[1] by using the already published structure from the recombinant 6aJL2 germline IGLV6-57 sequence (PDB: 2W0K). The germline sequence was computer-mutated to obtain the sequence from the λS-VL and the resulting structures were analyzed at physiological pH (7.5) at an interaction distance of 5 Å.

## Reporting summary

Further information on research design is available in the Nature Portfolio Reporting Summary linked to this article.

## Data availability

Raw data from RNA-seq have been deposited in the European Nucleotide Archive database under access code PRJE81509 (https://www.ebi.ac.uk/ena/browser/view/PRJEB81509) and Gene expression Omnibus (GEO) database under access code GSE286368. λS-LC sequence has been deposited in GenBank with the accession code PQ628092. The mass spectrometry data on purified amyloid fibrils have been deposited to the ProteomeXchange Consortium via the PRIDE [1] partner repository with the dataset identifier PXD059000 (purified amyloid fibrils) (https://proteomecentral.proteomexchange.org/cgi/GetDataset?ID=PXD059000) and PXD059046 (proteolytic patterns) (https://proteomecentral.proteomexchange.org/cgi/GetDataset?ID=PXD059046). All unique biological materials (recombinant patient's light chains, transgenic mice) are available from the corresponding author. Source data are provided with this paper.

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

## Acknowledgements

The authors thank the staff of the Biologie Intégrative Santé Chimie Environnement (BISCEm) technical platforms at the University of Limoges (animal, cell cytometry and microscopy transgenesis facilities), the Department of Pathology of Poitiers. The authors also thank the GeT-Santé facility (I2MC, Inserm, Génome et Transcriptome, GenoToul, Toulouse, France), Pr. Jean Feuillard for bioinformatic tips and Dr. Nikolaou Panagiota-Efstathia and Dr. Pauline Caillard for their helpful advice on NT-proBNP kit and cardiac ultrasound analysis respectively. This work was supported by grants from Agence National de la Recherche (#ANR-21-CE17-0040-01), Fondation pour la Recherche Médicale (# FRM-EQU202203014615), Fondation Française pour la recherche sur le Myelome et les Gammapathies monoclonales (FFRMG) and Ligue nationale contre le cancer. G.M.-R. is funded by fellowships from Région Nouvelle Aquitaine and Société Française d'Hématologie. M.V.A. was funded by fellowship from Fondation ARC pour la Recherche sur le Cancer. A.L. is funded by a grant from ERA4Health partnership (Cardinnov #ANR-23-R4HC-0001-03), S.B. is supported by Center Hospitalier Universitaire Dupuytren Limoges and Plan National Maladies Rares. G.R.C. is funded by Fondation pour la Recherche Médicale (# FRM-EQU202203014615). K.W.S. is funded by Agence National de la Recherche (#ANR-21-CE17-0040-01). M.E. was supported by Deutsche Forschungsgemeinschaft by grant EH 100/21-1. A.C. was supported by Deutsche Forschungsgemeinschaft by grant CA 2420/2-1. F.L. was supported by Cariplo-Telethon Joint Call for Applications # 2022-0578 and Grant Fondo Italiano per la Scienza (FIS 2021) # FIS00001548.

## Author contributions

Conceptualization: G.M.-R., M.V.A, and C.S. Methodology and Analysis: G.M.-R., M.V.A, S.B., G.R.C., K.W.S., A.L., L.P., M.M., P.S., E.P., L.R., F.L., D.C., A.R., S.K., C.Or., C.Ob., A.B. and C.S. Data Visualization and interpretation and general advice: M.R., V.J., A.J., E.N., A.C., S.G., L.D., and M.E. Bioinformatic analysis: J.P., G.M-R., and C.S. Original draft: G.M.-R., M.V.A., and C.S. Funding acquisition and Resources: C.S. Review and Correction by G.M.-R., S.B., P.S., E.P., F.L., D.C., M.E., F.B., and C.S. All authors read and approved the final manuscript.

## Competing interests

The authors declare no competing interests.
