## [Peer Review File · Nature Communications]

A mouse model of cardiac immunoglobulin light chain amyloidosis reveals insights into tissue accumulation and toxicity of amyloid fibrils

Corresponding Author: Professor Christophe Sirac

Version 0:

Reviewer comments:

Reviewer #1

(Remarks to the Author)

The authors report the generation and characterization of a transgenic mouse model of AL amyloidosis, a type of systemic amyloidosis experienced by patients who have plasma cell dyscrasias producing free amyloidogenic immunoglobulin light chains (the exact sequence of which is unique to each patient). These free light chains aggregate in a highly ordered fashion and become deposited in the extracellular space, disrupting organ function. In AL amyloidosis, the key organs affected are the heart and the kidney, and without early diagnosis, prompt and successful treatment, survival is poor. There are existing *in vivo* “models” of AL amyloidosis, but they are all non-physiological and have major drawbacks. There has long been a need for reproducible physiological models of AL amyloidosis.

Having first cloned an amyloidogenic immunoglobulin light chain cDNA from a patient with AL amyloidosis, they generated transgenic mice which express this free light chain at high concentration. This required a sophisticated experimental approach which the authors developed and previously reported in the context of other models.

In the absence of any further treatment, a small minority of transgenic mice expressing the free light chain developed cardiac amyloid deposits. In order to create a more consistent model, the authors sought to use the well-established approach of precipitating deposition by seeding with pre-formed amyloid fibrils. Following injection of amyloid seeds, amyloid deposition was massively accelerated, such that deposits were detected as early as 48h after seeding in some mice, with all mice developing cardiac amyloidosis within 6 months of seeding. Importantly the authors present evidence that the tissue amyloid contained the transgenically expressed light chain, and were not solely comprised of the injected seeds. The authors show various lines of evidence that the deposited material is genuine AL amyloid and, importantly, exclude AA and AApoAll amyloidosis, which also occur in mice. Phenotypic characterisation of the amyloidotic mice also included biomarker analysis and echocardiography, showing evidence of cardiac dysfunction, and gene expression profiling which showed a pro-fibrotic pattern. The findings are convincing, and represent an important advance.

The material used for seeding was recombinant variable region of the same amyloidogenic light chain that was expressed in the mice. The recombinant light chain fragment was incubated with shaking for a prolonged period to induce conversion into amyloid, and sonicated prior to injection. Full-length recombinant light chain did not form amyloid *in vitro* under similar conditions, adding to the existing evidence that cleavage of the precursor protein is important for amyloid formation, and not an epiphenomenon.

The authors were also able to seed amyloid deposition by injection of the “soluble amyloidogenic fragment” of the light chain, albeit with a long delay. It has been shown in mouse models of AA amyloidosis that vanishingly small amounts of AA amyloid seeds are sufficient to promote amyloid deposition (e.g. Baltz et al (1986) Isolation and characterisation of amyloid enhancing factor (AEF) in Amyloidosis (eds Glenner et al, Plenum Press)). The authors used several assays to look for aggregated protein, and failed to detect any aggregates. The precedent for seeding of AA amyloid with tiny amounts of pre-formed amyloid means that the authors must caveat this finding with the possibility that the injected material may have contained very small amount of pre-formed amyloid below the limit of detection.

One finding which at face value was unexpected was the absence of serum amyloid P component (SAP) in the amyloid.

This protein is a universal component of amyloid deposits and, in proteomic analysis, it is used as part of an amyloid signature. Binding of SAP to amyloid is calcium dependent. The authors report that preparation of the amyloid fibrils for the proteomic analysis involved washing with EDTA-containing buffer, followed by several water extraction steps. These treatments would result in the dissociation of SAP from the amyloid. The absence of SAP is likely an artefactual result.

Overall, this is an important piece of work which represents a major advance in the field. Because of the uniqueness of each patient's light chains, the model reported is of one specific AL amyloidosis. There is a case to be made for further models of AL amyloidosis to be made (e.g. expressing light chains from patients with no cardiac disease), and this paper provides a blueprint for such endeavours.

Minor points:

The exact full -length sequences of the λ S-LC and r λ S-VL should be reported, and the sequence identity of λ S-LC and r λ S-LC should be confirmed.

Page 8 line 3: exogenous seeding of AA amyloid deposition is not necessary in animal models (seeding synchronises onset of and accelerates deposition, but is not essential).

Page 9 line 13: the data discussed show that the λ S-LCs are an abundant constituent of the amyloid fibrils, but do not show that they are the main constituent.

Various: The usual abbreviation for extracellular matrix is ECM; use of the abbreviation EM (typically used for electron microscopy) is confusing.

Page 13 penultimate line: Fig. S7O should be Fig. S6O

Page 15 line 22: The statement "This means that once formed, these oligomers quickly become resistant to proteolysis." is an overinterpretation.

Page 16 line 4: plasma, not serum

Fig 1 legend penultimate line: > 6 months, not < 6 months

Some of the figures labelling is unclear (e.g. 3D, 5B).

Some figure legends do not clearly state what is shown in the figure.

Reviewer #2

(Remarks to the Author)

The manuscript by Martinez-Rivas describes the development of a new mouse model of AL amyloidosis, which exhibits many of the same pathological features as the human condition. The model shows that AL amyloidosis itself had limited toxic effects on tissues, but toxicity could be induced with administration of amyloid fibrils made from the variable domain. The model has provided some important insights into amyloid formation and the factors contributing to tissue toxicity. While the validity of the model in describing human disease pathogenesis will require additional work, the model will be a valuable asset for the field for pre-clinical therapy development.

The studies are conducted to a high overall standard. Here are some considerations aimed at improve some minor aspects of the manuscript:

- Can the authors speculate on why the mouse model didn't develop kidney amyloidosis, unlike the patient?
- Fig 3G, which unlike previous images in this panel, is not characterizing amyloidosis, would benefit from comparison to control images
- The ejection fraction of all mice (in both groups) appears compromised to near heart failure levels. This has implications for the validity of the echo analysis. Can the authors comment?
- It would be worth noting in the results section that with in diastolic dysfunction with increased filling pressures, differences in E:A would be unexpected.

Minor:

- Figure 5D: relabel middle panel as Spleen.
- Can heart rate information be added to the echocardiography methods (ie. What was HR range used).

Reviewer #3

(Remarks to the Author)

The manuscript, "A mouse model of cardiac AL amyloidosis unveils mechanisms of tissue accumulation and toxicity of amyloid fibrils" reports a well-designed and well-executed study to establish a transgenic mouse model of cardiac AL amyloidosis. The study also provides valuable insights into the mechanisms of light chain amyloid deposition and cardiomyocyte toxicity.

This manuscript is the first to present a mouse model of cardiac AL amyloidosis in which AL deposition is associated with pathological changes in heart anatomy and myocardial function, including an increase in cardiac biomarkers such as NT-

proBNP, indicative of cardiac damage. Developing an animal model that replicates AL deposition and subsequent tissue injury has been a longstanding goal for over two decades. This study is especially timely, as structural techniques such as solid-state nuclear magnetic resonance and cryo-electron microscopy have significantly advanced our knowledge of the structure of AL fibrils. Despite these technological strides, how light chain amyloid deposition occurs *in vivo* remains unexplored mainly due to the absence of an appropriate animal model. Therefore, this study has the potential to significantly enhance research on the mechanisms of light chain amyloid aggregation and the resulting pathophysiological events. This is entirely original research, reflecting years of focused work from the corresponding author's laboratory in collaboration with other experts in the field.

This study has several significant findings. One is the tissue deposition pattern similarity in animals administered with seeds and those injected with the VL domain. This finding suggests that the amyloid deposition in the heart in animals that were administered seeds was not due to the preferential localization of these aggregates in the organ but to the tissue tropism of the circulating full-length light chain. Hence, this model can help to understand the molecular basis of organ tropism that distinguished AL deposition. Another relevant finding is the similarity in the fragmentation pattern of the full-length light chain of the AL deposits in the mouse with that observed in a patient with AL amyloidosis. This finding suggests a remarkable similarity in the natural mechanisms of removal of AL deposits in humans and mice. This finding makes this model suitable for research aimed at developing strategies that promote the removal of AL deposits by activating tissue mechanisms, such as proteolysis and macrophage-mediated phagocytosis.

Another significant finding of this study is the demonstration of spontaneous AL deposition after several months in mice expressing the light chain but not given seeds or VL. This demonstrates that the full-length light chain can form AL deposits in the mouse if one waits long enough. Obviously, for practical purposes, an animal model of AL amyloidosis must reproduce the key pathological events of the disease within a short time after the animal's birth. However, evidence suggests that, depending on the biophysical and structural properties of the monoclonal light chain and the influence of tissue microenvironmental factors, AL deposition may be a slow process. This appears to be true in the mouse as well. The study is methodologically robust, utilizing a comprehensive array of biochemical, physiological, and molecular techniques to characterize amyloid deposits and assess tissue damage. Overall, the work meets the expected standards of rigor in this field.

Questions and comments

The serum FLC level in λ S-DH double homozygous mice was fourfold higher than in the λ S-PT patient at diagnosis. Unlike patient λ S-PT, whose kidneys were the main target of AL deposition, λ S-DH mice presented mainly cardiac amyloidosis. Could the higher level of circulating FLC determine the difference in LC organ tropism?

The authors state that λ S-VL rapidly started fibrillogenesis *in vitro*, as observed with the increase of ThT fluorescence, starting at two days under aggregating conditions. Based on this finding, the authors state that λ S-VL is highly amyloidogenic. It is well-known that the kinetic of *in vitro* fibrillogenesis of the light chains is method-dependent. However, compared with what has been reported for other λ 6 rVLs, including Wil, Jto, and germline 6aJL2, the kinetics of *in vitro* fibrillogenesis of the protein λ S-VL is slow.[1-4] In this study, the aggregation experiment aimed to demonstrate that the rVL is significantly more fibrillogenic than the full-length chain, partly explaining why mice did not develop AL deposition without seeding. However, it cannot be excluded that the absence of AL deposition is also a consequence of having chosen an amyloidogenic λ 6 light chain to produce the transgenic model with relatively low aggregation propensity. Could it have been Wil or even the germline 6aJL2 better candidates? Including some of these proteins in the *in vitro* fibrillogenesis experiments as a reference would have provided relevant information regarding the intrinsic aggregation propensity of the λ S light chain.

Proteomics of AL amyloid extracted from cardiac deposits showed the absence of the serum amyloid component P (SAP). This amyloid-associated molecule is thought to play an essential role in the stability of amyloid fibrils. Is the inability of mouse SAP to interact with human amyloidogenic light chains related to the resistance of mice to human amyloidogenic light chain deposition?

In the Discussion, the authors state that "based on our findings, it is tempting to argue that a similar partial proteolysis of fLCs, releasing fragments containing destabilized VL that can rapidly form the first protease-resistant nuclei, occurs in serum or extracellular matrix of organs and may initiate the elongation of amyloid fibrils with the circulating monoclonal full-length LC." In support of this statement, the authors mention the recently published study conducted by Francesca Lavatelli et al, entitled "Truncation of the constant domain drives amyloid formation by immunoglobulin light chains." [5] In this study, Lavatelli et al. characterized the folding stability and aggregation propensity of proteolytic fragments of the amyloidogenic λ 6 light chain AL55." It is essential to clarify that the proteolytic fragments of AL55 studied by Francesca Lavatelli et al. were extracted from *ex-vivo* AL deposits. The proteomic analysis of AL deposits in patient AL55 revealed a fragmentation pattern that was more in agreement with post-fibrillogenic proteolysis. Therefore, although the study performed by Lavatelli et al. adds to our knowledge about the role of the interactions between VL and CL in the mechanism of light chain amyloidosis, it does not provide evidence that the proteolytic fragments of AL55 were produced before the assembling of the light chain into the amyloid fibril. This manuscript also does not provide compelling evidence of full-length light chain proteolysis preceding amyloid aggregation. What this study does demonstrate is that VL can seed tissue deposition of the full-length light chain. The problem with the hypothesis of the proteolysis that releases an amyloidogenic VL as the triggering event of light chain amyloid aggregation is that the VL tends to be more susceptible to proteolytic cleavage than the CL due to its lower folding stability and greater conformational flexibility. Since the VL is the aggregation-prone domain, it can be anticipated that it forms the core not only of the amyloid fibrils but also of the heterogeneous ensemble of oligomers and prefibrillar aggregates that populate the aggregation pathway. Therefore, an alternative interpretation for the fragmentation pattern observed in this study is that the VL is protected early from proteolysis in the aggregation pathway, whereas the CL, exposed at the periphery of AL oligomers and fibrils, is proteolyzed. This is consistent with the significant overexpression of several extracellular metalloproteinases and cathepsins in the extracellular space that accompanied AL deposition in mice seeded with preformed fibrils and VL, changes that were not observed in mice expressing the full-length light chain that were not seeded. This suggests that AL deposition is the activating factor for metalloproteinases and cathepsins overexpression. Finally, the hallmark of immunoglobulin light chains is their sequence diversity, resulting in unique disease presentations in

AL amyloidosis patients. As such, no single animal model can fully replicate the heterogeneity of human AL amyloidosis. The authors should briefly address this limitation in their discussion.

1. Wall J, Schell M, Murphy C, Hrcic R, Stevens FJ, Solomon A. Thermodynamic instability of human lambda 6 light chains: correlation with fibrillogenicity. *Biochemistry*. Oct 19 1999;38(42):14101-8. doi:10.1021/bi991131j
2. del Pozo Yauner L, Ortiz E, Sanchez R, et al. Influence of the germline sequence on the thermodynamic stability and fibrillogenicity of human lambda 6 light chains. *Proteins*. Aug 2008;72(2):684-92. doi:10.1002/prot.21934
3. Blancas-Mejia LM, Tellez LA, del Pozo-Yauner L, Becerril B, Sanchez-Ruiz JM, Fernandez-Velasco DA. Thermodynamic and kinetic characterization of a germ line human lambda6 light-chain protein: the relation between unfolding and fibrillogenesis. *J Mol Biol*. Mar 6 2009;386(4):1153-66. doi:10.1016/j.jmb.2008.12.069
4. del Pozo-Yauner L, Wall JS, Gonzalez Andrade M, et al. The N-terminal strand modulates immunoglobulin light chain fibrillogenesis. *Biochem Biophys Res Commun*. Jan 10 2014;443(2):495-9. doi:10.1016/j.bbrc.2013.11.123
5. Lavatelli F, Natalello A, Marchese L, et al. Truncation of the constant domain drives amyloid formation by immunoglobulin light chains. *J Biol Chem*. Apr 2024;300(4):107174. doi:10.1016/j.jbc.2024.107174

Version 1:

Reviewer comments:

Reviewer #1

(Remarks to the Author)

I am fully satisfied with the authors' responses to the issues I raised in my first review.

Reviewer #2

(Remarks to the Author)

Thank you for clarifying the points raised for the additional data where requested.

Reviewer #3

(Remarks to the Author)

The authors have thoroughly addressed the reviewer's questions, and the changes introduced in the revised version effectively address the reviewer's suggestions and concerns. The discussion of the results is now more precise, with a more accurate interpretation that considers alternative perspectives. Overall, the revisions improve the quality of the manuscript, which, as noted in the original review, represents a significant contribution to understanding the mechanism of amyloid light chain aggregation in vivo and the pathogenic processes triggered by this event.

Open Access This Peer Review File is licensed under a Creative Commons Attribution 4.0 International License, which permits use, sharing, adaptation, distribution and reproduction in any medium or format, as long as you give appropriate credit to the original author(s) and the source, provide a link to the Creative Commons license, and indicate if changes were

made.

First, we would like to sincerely thank the three reviewers for their positive feedback on our work and their constructive comments, which should greatly improve the quality of our manuscript.

Please find below our point-by-point response to reviewers (in blue)

REVIEWER COMMENTS

Reviewer #1 (Remarks to the Author):

The authors report the generation and characterization of a transgenic mouse model of AL amyloidosis, a type of systemic amyloidosis experienced by patients who have plasma cell dyscrasias producing free amyloidogenic immunoglobulin light chains (the exact sequence of which is unique to each patient). These free light chains aggregate in a highly ordered fashion and become deposited in the extracellular space, disrupting organ function. In AL amyloidosis, the key organs affected are the heart and the kidney, and without early diagnosis, prompt and successful treatment, survival is poor. There are existing in vivo “models” of AL amyloidosis, but they are all non-physiological and have major drawbacks. There has long been a need for reproducible physiological models of AL amyloidosis.

Having first cloned an amyloidogenic immunoglobulin light chain cDNA from a patient with AL amyloidosis, they generated transgenic mice which express this free light chain at high concentration. This required a sophisticated experimental approach which the authors developed and previously reported in the context of other models.

In the absence of any further treatment, a small minority of transgenic mice expressing the free light chain developed cardiac amyloid deposits. In order to create a more consistent model, the authors sought to use the well-established approach of precipitating deposition by seeding with pre-formed amyloid fibrils. Following injection of amyloid seeds, amyloid deposition was massively accelerated, such that deposits were detected as early as 48h after seeding in some mice, with all mice developing cardiac amyloidosis within 6 months of seeding. Importantly the authors present evidence that the tissue amyloid contained the transgenically expressed light chain, and were not solely comprised of the injected seeds. The authors show various lines of evidence that the deposited material is genuine AL amyloid and, importantly, exclude AA and AApoAll amyloidosis, which also occur in mice. Phenotypic characterisation of the amyloidotic mice also included biomarker analysis and echocardiography, showing evidence of cardiac dysfunction, and gene expression profiling which showed a pro-fibrotic pattern. The findings are convincing, and represent an important advance.

The material used for seeding was recombinant variable region of the same amyloidogenic light chain that was expressed in the mice. The recombinant light chain fragment was incubated with shaking for a prolonged period to induce conversion into amyloid, and sonicated prior to injection. Full-length recombinant light chain did not form amyloid in vitro

under similar conditions, adding to the existing evidence that cleavage of the precursor protein is important for amyloid formation, and not an epiphenomenon.

The authors were also able to seed amyloid deposition by injection of the “soluble amyloidogenic fragment” of the light chain, albeit with a long delay. It has been shown in mouse models of AA amyloidosis that vanishingly small amounts of AA amyloid seeds are sufficient to promote amyloid deposition (e.g. Baltz et al (1986) Isolation and characterisation of amyloid enhancing factor (AEF) in Amyloidosis (eds Glenner et al, Plenum Press)). The authors used several assays to look for aggregated protein, and failed to detect any aggregates. The precedent for seeding of AA amyloid with tiny amounts of pre-formed amyloid means that the authors must caveat this finding with the possibility that the injected material may have contained very small amount of pre-formed amyloid below the limit of detection.

We now added a sentence to state this possibility in the discussion (p.15: “Although we cannot formally exclude that the injected material may have contained tiny amount of pre-formed amyloid seeds below the limit of detection by HPLC, this result means that once formed, these oligomers persist for a long period of time in tissues.”)

One finding which at face value was unexpected was the absence of serum amyloid P component (SAP) in the amyloid. This protein is a universal component of amyloid deposits and, in proteomic analysis, it is used as part of an amyloid signature. Binding of SAP to amyloid is calcium dependent. The authors report that preparation of the amyloid fibrils for the proteomic analysis involved washing with EDTA-containing buffer, followed by several water extraction steps. These treatments would result in the dissociation of SAP from the amyloid. The absence of SAP is likely an artefactual result.

We were also confused by this result at first and we agree with the reviewer that the absence of SAP in fibril extracts could be due to the use of EDTA during the protocol. But we also failed to detect significant SAP deposits in amyloid tissues (not treated with EDTA) using IF studies with an anti-mouse SAP antibody (R&D system AF2558) (Fig.S5B). In fact, although SAP is a well-known amyloid signature protein in humans, to our knowledge, it is less clear in mice either by IF or MS. For instance, it was not detected by MS on microdissections in a human transgenic AApoA2 model (PMID: 31200944) and detected only in 4/7 microdissections in a human transgenic ATTR model with quite low mascot scores (PMID: 34876572) which is in sharp contrast with the high and constant detection of SAP in human samples.

For the purpose of the present revisions, we performed new MS analysis on extracted fibrils from 5 AL positive mice (3 extracts) (new Fig.6B): SAP remains barely detectable with only one sample out of 3 with a significant (but still low) EmPAI score.

We now discuss in the results and discussion sections several possibilities for the absence or low level of SAP:

- 1- *In mice, SAP is the main acute phase protein but its basal level is very low as compared to human in non-inflammatory mice, especially in C57/bl6 mice. Since we are in a*

mixed genetic background, we performed ELISA to know the level in the AL mice and found a mean level of 1.9 µg/mL, which is more than 10 times lower than the human level (**new Fig.S5C**).

- 2- It is known from previous studies that the avidity of mouse SAP for amyloidosis is lower than that of human SAP which could also explain the absence of SAP.

(p.10: "Serum Amyloid P component (SAP) was barely detected in only 1/3 extracts (Fig. 6B). Accordingly, we were not able to detect SAP in cardiac tissues of amyloid positive mice by immunofluorescence (Fig. S5B) which could be due to the low level of circulating mouse SAP in our mice (Fig. S5C, 1.92 µg/ml ± 0.65, mean ± SEM, n=12) compared to human (~ 25 µg/mL)⁴².")

(p15:"...except SAP which was barely detectable by MS and not detected by IF. This contrasts with previous models of ATTR amyloidosis^{50,37} although it may be explained by the low level of circulating SAP in our mice and the weak avidity of mouse SAP to amyloid deposits⁵¹")

Since SAP was reported to protect fibrils from degradation, we now discuss also the possibility that the low level of SAP in our mice could explain the absence of spontaneous amyloid deposition

(p.17: "...the absence of associated serum amyloid P component to the fibrils, know to protect fibrils from degradation⁴¹, could also explain this observation").

Overall, this is an important piece of work which represents a major advance in the field. Because of the uniqueness of each patient's light chains, the model reported is of one specific AL amyloidosis. There is a case to be made for further models of AL amyloidosis to be made (e.g. expressing light chains from patients with no cardiac disease), and this paper provides a blueprint for such endeavours.

We fully agree and other models (with renal only involvement) are on their way in our lab *(p.20: "Other models using the same strategy but producing LCs from patients with different organ involvement need to be developed to better represent and understand the diversity of this disease.")*.

Minor points:

The exact full-length sequences of the λS-LC and rλS-VL should be reported, and the sequence identity of λS-LC and rλS-LC should be confirmed.

We have now added the full-length sequences with the alignment of λS-LC, rλS-LC and rλS-VL in Fig. S2A. We have now indicated that rλS-VL contains 2 extra AA in N-term in the supplemental methods. We apologize for not mentioning this on the initial manuscript.

Page 8 line 3: exogenous seeding of AA amyloid deposition is not necessary in animal models (seeding synchronises onset of and accelerates deposition, but is not essential).

We agree and apologize for this unfortunate shortcut, seeding only accelerating deposits in AA models. We modified the text accordingly.

Page 9 line 13: the data discussed show that the λ S-LCs are an abundant constituent of the amyloid fibrils, but do not show that they are the main constituent.

Once again we agree and thank the reviewer for his accurate reading of our work. We modified the text accordingly.

Various: The usual abbreviation for extracellular matrix is ECM; use of the abbreviation EM (typically used for electron microscopy) is confusing.

ECM for extracellular matrix is used throughout the text now.

Page 13 penultimate line: Fig. S7O should be Fig. S6O

Done

Page 15 line 22: The statement "This means that once formed, these oligomers quickly become resistant to proteolysis." is an overinterpretation.

We modified the text to tone down this statement.

Page 16 line 4: plasma, not serum

Done

Fig 1 legend penultimate line: > 6 months, not < 6 months

Done

Some of the figures labelling is unclear (e.g. 3D, 5B).

We have reviewed the text and hope we have not left inaccuracies.

Some figure legends do not clearly state what is shown in the figure.

We have now tried to clarify the content of the figures in the legends.

Reviewer #2 (Remarks to the Author):

The manuscript by Martinez-Rivas describes the development of a new mouse model of AL amyloidosis, which exhibits many of the same pathological features as the human condition. The model shows that AL amyloidosis itself had limited toxic effects on tissues, but toxicity could be induced with administration of amyloid fibrils made from the variable domain.

The model has provided some important insights into amyloid formation and the factors contributing to tissue toxicity. While the validity of the model in describing human disease pathogenesis will require additional work, the model will be a valuable asset for the field for pre-clinical therapy development.

The studies are conducted to a high overall standard. Here are some considerations aimed at improve some minor aspects of the manuscript:

- Can the authors speculate on why the mouse model didn't develop kidney amyloidosis, unlike the patient?

They actually developed amyloid deposits in kidney but did not reach significant levels in the timeframe of our experiments. This is indeed an intriguing point that we now discuss more into details in the discussion (p.17: *"One could speculate that the tropism for kidney of the LC or the injected seeds was low because of missing receptors that could bind and retain them into glomeruli or the mouse kidneys express factors that protect them from amyloid formation (efficient extracellular proteostasis). But we cannot exclude that technical issues like the i.v. route for seeds and VL injections or the higher level of fLC in the mice compared to human could also, for unknown reasons, favored their early deposition in heart at the expense of the kidney or other organs."*)

- Fig 3G, which unlike previous images in this panel, is not characterizing amyloidosis, would benefit from comparison to control images

We have now added a control heart for comparison (**new Fig 3G**)

- The ejection fraction of all mice (in both groups) appears compromised to near heart failure levels. This has implications for the validity of the echo analysis. Can the authors comment?

We evaluated cardiac anatomy and function in animals older than 14 months. It is well known that mice exhibit an age-related decline in ejection fraction but in our study, the ejection fraction values observed are consistent with those reported in the literature for mice older than 12 months (Fig. 1L, [10.1038/s43587-024-00612-4; https://www.ncbi.nlm.nih.gov/pmc/articles/PMC5807035](https://www.ncbi.nlm.nih.gov/pmc/articles/PMC5807035)). This physiological change is not considered indicative of heart failure with systolic dysfunction, which is typically defined as an ejection fraction below 45%. We now indicate in the results that EF slightly below 60% are consistent with aged mice (p.12: *"Of note, ventricular ejection fraction values were slightly below 60% in both groups which is consistent with the progressive age-related decline observed in mice older than 12 months and not indicative of heart failure with systolic dysfunction⁴⁶."*)

As indicated in the methods, we performed echography following the American Physiological Society guidelines for cardiac measurements in mice. We used a constant isoflurane concentration of 2% and strictly controlled body temperature ($37^{\circ}\text{C} \pm 0.4$) throughout the 10-minute scan. This protocol ensured a stable heart rate of over 400 bpm, with no significant difference between the two groups (477 ± 70.0 vs. 440 ± 30 bpm, Mean \pm SD, ns).

- It would be worth noting in the results section that with in diastolic dysfunction with increased filling pressures, differences in E:A would be unexpected.

Although the overall E/A ratio difference was not statistically significant, individual variability may indicate distinctions between pseudo-normal and restrictive diastolic dysfunction. The limited number of animals used in this study makes it difficult to definitively categorize the type of diastolic dysfunction observed. However, since there was no significant differences and such differences would not be expected, we have now removed the E/A results from the text to focus only on the GLS, which should have been different between the 2 groups (*p*12-13: “Differences in the global longitudinal strain (GLS) was not significant between groups (*p*=0.7879) (Fig. S6G) but a reduced ventricular filling capacity was observed by the increase in left atrial (LA) volume and a decrease in LA ejection fraction (Fig. S6H).”). We hope this change is acceptable.

Minor:

- Figure 5D: relabel middle panel as Spleen.

Done

- Can heart rate information be added to the echocardiography methods (ie. What was HR range used).

We have now indicated the heart rate data (new Fig.S6D)

Reviewer #3 (Remarks to the Author):

The manuscript, "A mouse model of cardiac AL amyloidosis unveils mechanisms of tissue accumulation and toxicity of amyloid fibrils" reports a well-designed and well-executed study to establish a transgenic mouse model of cardiac AL amyloidosis. The study also provides valuable insights into the mechanisms of light chain amyloid deposition and cardiomyocyte toxicity.

This manuscript is the first to present a mouse model of cardiac AL amyloidosis in which AL deposition is associated with pathological changes in heart anatomy and myocardial function, including an increase in cardiac biomarkers such as NT-proBNP, indicative of cardiac damage. Developing an animal model that replicates AL deposition and subsequent tissue injury has been a longstanding goal for over two decades. This study is especially timely, as structural techniques such as solid-state nuclear magnetic resonance and cryo-electron microscopy have significantly advanced our knowledge of the structure of AL fibrils. Despite these technological strides, how light chain amyloid deposition occurs *in vivo* remains unexplored mainly due to the absence of an appropriate animal model. Therefore, this study has the potential to significantly enhance research on the mechanisms of light chain amyloid aggregation and the resulting pathophysiological events.

This is entirely original research, reflecting years of focused work from the corresponding author's laboratory in collaboration with other experts in the field.

This study has several significant findings. One is the tissue deposition pattern similarity in animals administered with seeds and those injected with the VL domain. This finding suggests

that the amyloid deposition in the heart in animals that were administered seeds was not due to the preferential localization of these aggregates in the organ but to the tissue tropism of the circulating full-length light chain. Hence, this model can help to understand the molecular basis of organ tropism that distinguished AL deposition. Another relevant finding is the similarity in the fragmentation pattern of the full-length light chain of the AL deposits in the mouse with that observed in a patient with AL amyloidosis. This finding suggests a remarkable similarity in the natural mechanisms of removal of AL deposits in humans and mice. This finding makes this model suitable for research aimed at developing strategies that promote the removal of AL deposits by activating tissue mechanisms, such as proteolysis and macrophage-mediated phagocytosis.

Another significant finding of this study is the demonstration of spontaneous AL deposition after several months in mice expressing the light chain but not given seeds or VL. This demonstrates that the full-length light chain can form AL deposits in the mouse if one waits long enough. Obviously, for practical purposes, an animal model of AL amyloidosis must reproduce the key pathological events of the disease within a short time after the animal's birth. However, evidence suggests that, depending on the biophysical and structural properties of the monoclonal light chain and the influence of tissue microenvironmental factors, AL deposition may be a slow process. This appears to be true in the mouse as well.

The study is methodologically robust, utilizing a comprehensive array of biochemical, physiological, and molecular techniques to characterize amyloid deposits and assess tissue damage. Overall, the work meets the expected standards of rigor in this field.

Questions and comments

The serum FLC level in λ S-DH double homozygous mice was fourfold higher than in the λ S-PT patient at diagnosis. Unlike patient λ S-PT, whose kidneys were the main target of AL deposition, λ S-DH mice presented mainly cardiac amyloidosis. Could the higher level of circulating FLC determine the difference in LC organ tropism?

In fact, the patient was first admitted in a nephrology department for a chronic kidney disease but at the time of diagnosis, he already presented with significant cardiac dysfunction (septum 14mm, diastolic dysfunction) However, it is true that mice develop less kidney deposits than cardiac. We cannot exclude the role of the LC level but at the same time, we do not have any argument that could be in line with this hypothesis. The kidney, with its filtration role, should be even more exposed to LC aggregation with a higher FLC level. We now discuss more into details different hypotheses that could explain this discrepancy with the patient (p.17: *“One could speculate that the tropism for kidney of the LC or the injected seeds was low because of missing receptors that could bind and retain them into glomeruli or the mouse kidneys express factors that protect them from amyloid formation (efficient extracellular proteostasis). But we cannot exclude that technical issues like the i.v. route for seeds and VL injections or the higher level of fLC in the mice compared to human could also, for unknown reasons, favored their early deposition in heart at the expense of the kidney or other organs.”*)

The authors state that r λ S-VL rapidly started fibrillogenesis in vitro, as observed with the increase of ThT fluorescence, starting at two days under aggregating conditions. Based on this

finding, the authors state that r λ S-VL is highly amyloidogenic. It is well-known that the kinetic of *in vitro* fibrillogenesis of the light chains is method-dependent. However, compared with what has been reported for other lambda-6 rVLs, including Wil, Jto, and germline 6aJL2, the kinetics of *in vitro* fibrillogenesis of the protein r λ S-VL is slow.[1-4] In this study, the aggregation experiment aimed to demonstrate that the rVL is significantly more fibrillogenic than the full-light chain, partly explaining why mice did not develop AL deposition without seeding. However, it cannot be excluded that the absence of AL deposition is also a consequence of having chosen an amyloidogenic lambda-6 light chain to produce the transgenic model with relatively low aggregation propensity. Could it have been Wil or even the germline 6aJL2 better candidates? Including some of these proteins in the *in vitro* fibrillogenesis experiments as a reference would have provided relevant information regarding the intrinsic aggregation propensity of the λ S light chain.

This is a very interesting comment. We agree that the Variable λ S domain seems to have a mild aggregation propensity *in vitro* in comparison with other published VL6-57 proteins and that it could explain the absence of spontaneous amyloidosis. We now discuss this hypothesis in the discussion section (p.17: *“The low amyloidogenicity of the λ S-VL protein compared to other published IGLV6-57 domains^{35,55} ...could also explain this observation.”*) and we have toned down our message on the high amyloidogenicity of λ S in the results section (p.7: *“Accordingly, r λ S-VL started fibrillogenesis *in vitro* at 2 days under aggregating conditions as observed with the increase of ThT fluorescence”* and p.7: *“However, the λ S variable domain is able form fibrils *in vitro*.”*)

However, we do not think that *in vitro* assays reflect the propensity to form amyloidosis *in vivo*. Especially because most *in vitro* studies used the variable domain alone and not the full length LC as *in vivo*. As mentioned in the manuscript, our transgenic LC was extracted from a patient with overt AL and a low level of FLC, which, to our opinion, reflects its strong propensity to form amyloidosis *in vivo* from the full length LC. On the contrary, for instance, Jto, which seems to be more prone to form aggregates *in vitro* with the VL region only, was extracted from a patient with multiple myeloma and no sign of AL amyloidosis.

Proteomics of AL amyloid extracted from cardiac deposits showed the absence of the serum amyloid component P (SAP). This amyloid-associated molecule is thought to play an essential role in the stability of amyloid fibrils. Is the inability of mouse SAP to interact with human amyloidogenic light chains related to the resistance of mice to human amyloidogenic light chain deposition?

We totally agree and discuss this possibility in the discussion (p.17: *“The low amyloidogenicity of the λ S-VL protein compared to other published IGLV6-57 domains^{35,55} and the absence of associated serum amyloid P component to the fibrils, known to protect fibrils from degradation⁴², could also explain this observation.”*)

In the Discussion, the authors state that “based on our findings, it is tempting to argue that a similar partial proteolysis of fLCs, releasing fragments containing destabilized VL that can rapidly form the first protease-resistant nuclei, occurs in serum or extracellular matrix of

organs and may initiate the elongation of amyloid fibrils with the circulating monoclonal full-length LC.” In support of this statement, the authors mention the recently published study conducted by Francesca Lavatelli et al, entitled “Truncation of the constant domain drives amyloid formation by immunoglobulin light chains.”[5] In this study, Lavatelli et al. characterized the folding stability and aggregation propensity of proteolytic fragments of the amyloidogenic λ 6 light chain AL55.” It is essential to clarify that the proteolytic fragments of AL55 studied by Francesca Lavatelli et al. were extracted from ex-vivo AL deposits. The proteomic analysis of AL deposits in patient AL55 revealed a fragmentation pattern that was more in agreement with post-fibrillogenetic proteolysis. Therefore, although the study performed by Lavatelli et al. adds to our knowledge about the role of the interactions between VL and CL in the mechanism of light chain amyloidosis, it does not provide evidence that the proteolytic fragments of AL55 were produced before the assembling of the light chain into the amyloid fibril. This manuscript also does not provide compelling evidence of full-length light chain proteolysis preceding amyloid aggregation. What this study does demonstrate is that VL can seed tissue deposition of the full-length light chain. The problem with the hypothesis of the proteolysis that releases an amyloidogenic VL as the triggering event of light chain amyloid aggregation is that the VL tends to be more susceptible to proteolytic cleavage than the CL due to its lower folding stability and greater conformational flexibility. Since the VL is the aggregation-prone domain, it can be anticipated that it forms the core not only of the amyloid fibrils but also of the heterogeneous ensemble of oligomers and prefibrillar aggregates that populate the aggregation pathway. Therefore, an alternative interpretation for the fragmentation pattern observed in this study is that the VL is protected early from proteolysis in the aggregation pathway, whereas the CL, exposed at the periphery of AL oligomers and fibrils, is proteolyzed. This is consistent with the significant overexpression of several extracellular metalloproteinases and cathepsins in the extracellular space that accompanied AL deposition in mice seeded with preformed fibrils and VL, changes that were not observed in mice expressing the full-length light chain that were not seeded. This suggests that AL deposition is the activating factor for metalloproteinases and cathepsins overexpression.

This is a stimulating and yet unresolved debate on how amyloidosis starts. It is obvious, and we mention it in the introduction (p.3), that some proteolysis of the exposed unfolded CL occurs after the formation of fibrils. However, it does not exclude the possibility that some proteolysis also occur before and allow the initial conformational change of the LC required for the formation of the first seeds. This is the message of the last F.Lavatelli’s paper in JBC (talking on her own behalf since she is co-author of the present manuscript). She showed that the only way to convert a full length AL55 LC into amyloid fibrils is to mixed it with a truncated version of AL55.

Therefore, what we discuss in the manuscript is a hypothesis based on our data. The full length LC is produced at high level during months and months and do not form amyloidosis (except retrospectively in 2 mice that were not fully characterized) but a single injection of soluble VL, although its half-life is likely no more than few hours and, as you wisely point out, is highly susceptible to proteolytic degradation, is sufficient to trigger amyloidosis. Accordingly, we and others always failed to produce fibrils with full length LC in physiological conditions (i.e.

without low pH or high temperature) and all the publications cited by the reviewer used VL alone and not full length LC.

However, we understand the message of the reviewer and since we do not have any formal demonstration of our hypothesis, we have now toned down our hypothesis using terms that are more appropriate (p.16: *“it is tempting to hypothesize”* instead of *“argue”* and *“may rapidly form the first...”* instead *“can”*) and added the possibility of proteolysis after deposition in the discussion (p.16: *“However, in the absence of a formal experimental demonstration, we cannot exclude that in patients or in the few mice which developed spontaneous amyloidosis, the VL contained into the intact fLC was able to progressively acquire its amyloidogenic conformation, exposing the CL at the periphery of the fibrils, leading to their progressive degradation by activated resident proteases.”*)

Finally, the hallmark of immunoglobulin light chains is their sequence diversity, resulting in unique disease presentations in AL amyloidosis patients. As such, no single animal model can fully replicate the heterogeneity of human AL amyloidosis. The authors should briefly address this limitation in their discussion.

We agree once again and added a sentence about the importance of developing new models with different LCs to better reflect the heterogeneity of the disease (p.20: *“Other models using the same strategy but producing LCs from patients with different organ involvement need to be developed to better represent and understand the diversity of this disease.”*).

1. Wall J, Schell M, Murphy C, Hrcic R, Stevens FJ, Solomon A. Thermodynamic instability of human lambda 6 light chains: correlation with fibrillogenicity. *Biochemistry*. Oct 19 1999;38(42):14101-8. doi:10.1021/bi991131j
2. del Pozo Yauner L, Ortiz E, Sanchez R, et al. Influence of the germline sequence on the thermodynamic stability and fibrillogenicity of human lambda 6 light chains. *Proteins*. Aug 2008;72(2):684-92. doi:10.1002/prot.21934
3. Blancas-Mejia LM, Tellez LA, del Pozo-Yauner L, Becerril B, Sanchez-Ruiz JM, Fernandez-Velasco DA. Thermodynamic and kinetic characterization of a germ line human lambda6 light-chain protein: the relation between unfolding and fibrillogenesis. *J Mol Biol*. Mar 6 2009;386(4):1153-66. doi:10.1016/j.jmb.2008.12.069
4. del Pozo-Yauner L, Wall JS, Gonzalez Andrade M, et al. The N-terminal strand modulates immunoglobulin light chain fibrillogenesis. *Biochem Biophys Res Commun*. Jan 10 2014;443(2):495-9. doi:10.1016/j.bbrc.2013.11.123
5. Lavatelli F, Natalello A, Marchese L, et al. Truncation of the constant domain drives amyloid formation by immunoglobulin light chains. *J Biol Chem*. Apr 2024;300(4):107174. doi:10.1016/j.jbc.2024.107174